# Capturing Individual Human Preferences with Reward Features

**André Barreto**
Google DeepMind

**Vincent Dumoulin**
Google DeepMind

**Yiran Mao**
Google DeepMind

**Mark Rowland**
Google DeepMind

**Nicolas Perez-Nieves**
Google DeepMind

**Bobak Shahriari**
Google DeepMind

**Yann Dauphin**
Google DeepMind

**Doina Precup**
Google DeepMind

**Hugo Larochelle**
Google DeepMind

## Abstract

Reinforcement learning from human feedback usually models preferences using a reward function that does not distinguish between people. We argue that this is unlikely to be a good design choice in contexts with high potential for disagreement, like in the training of large language models. We formalise and analyse the problem of learning a reward model that can be specialised to a user. Using the principle of empirical risk minimisation, we derive a probably approximately correct (PAC) bound showing the dependency of the approximation error on the number of training examples, as usual, and also on the number of human raters who provided feedback on them. Based on our theoretical findings, we discuss how to best collect pairwise preference data and argue that adaptive reward models should be beneficial when there is considerable disagreement among users. We also propose a concrete architecture for an adaptive reward model. Our approach leverages the observation that individual preferences can be captured as a linear combination of a set of general reward features. We show how to learn such features and subsequently use them to quickly adapt the reward model to a specific individual, even if their preferences are not reflected in the training data. We present experiments with large language models illustrating our theoretical results and comparing the proposed architecture with a non-adaptive baseline. Consistent with our analysis, the benefits provided by our model increase with the number of raters and the heterogeneity of their preferences. We also show that our model compares favourably to adaptive counterparts, including those performing in-context personalisation.

## 1 Introduction

Reinforcement learning from human feedback (RLHF) can be useful when we do not have an explicit reward function but can distinguish between good and bad behaviour [15]. This is often the case in the training of large language models (LLMs), in which RLHF has been applied with great success [46, 48, 41, 58].

The most common way to collect human feedback is to ask humans to rank examples of an agent's behaviour. Usually, all the feedback is integrated to derive a single reward function that reflects as well as possible the preferences of the population of interest. Although this is a sensible strategy when preferences are homogeneous across the population, it may be less effective when there is considerable disagreement. This is likely the case in the training of LLMs.

39th Conference on Neural Information Processing Systems (NeurIPS 2025).

As an illustration, suppose that, given a pair of alternative responses to a subjective question, $51\%$ of the target audience prefer the first option while the remaining $49\%$ prefer the second. If we do not distinguish between users, we are left with two options: either we pick the preferred answer and leave $49\%$ of the users unhappy $100\%$ of the time, or we sample the answers proportionally to how often they are preferred and leave $100\%$ of the users unhappy approximately half of the time. Both solutions are clearly unsatisfactory.

We need reward models that can be specialised to users. Crucially, we want the model to be able to adapt to people outside of the group who provided feedback for training. In this paper we formalise and analyse the problem of learning a reward model with this ability. Using the principle of empirical risk minimisation, we derive a probably approximately correct (PAC) bound that shows how the approximation error depends not only on the number of training examples, but also on the number of human raters who provided feedback on them. This is an interesting problem from a theoretical standpoint because the data involved is not identically and independently distributed (i.i.d.). The resulting analysis provides a formal framework to discuss strategies for pairwise preference data collection and to assess the trade-offs associated with the use of an adaptive reward model.

We then turn our attention to the question of how to adapt a reward model to a new user in practice. Generalising to unseen users is challenging because it requires one to capture the subjective criteria that underlie human preferences—a non-trivial endeavour, as humans often cannot fully articulate the reasons why they prefer one behaviour over the other. A possible approach is to leverage precisely the type of pairwise preference data used in RLHF, since it allows the preference criteria to be indirectly elicited rather than explicitly spelled out. Building on our theoretical findings, we propose a simple, principled method to unveil the preference criteria underlying the RLHF data.

Our model leverages the fact that individual preferences can be captured as a linear combination of a set of general *reward features*. These features are unknown (in some cases to the person themself), so we need a systematic way to extract them from the data. We show how this can be accomplished using a very simple architecture and training. During the regular RLHF process, we use data coming from different individuals to learn common features that capture the preferences of the group. When the reward model is being specialised to an unknown user, the features are frozen, and only the coefficients of the linear combination must be learned. This results in a simple classification problem that can be reliably solved with a few training examples provided by the user.

## 2   Background

Our goal is to use human preference data to learn a reward function that can be used in RL to learn a policy or LLM [15, 56]. We will focus on and adopt the terminology from LLMs, although our analysis and proposed approach are more general. Let $\mathcal{T}$ be a finite set of symbols called *tokens*, let $\mathcal{X} := \cup_{i=0}^{l_x} \mathcal{T}^i$ be the *context space*, where $l_x$ is the maximum context length, and let $\mathcal{Y} := \cup_{i=0}^{l_y} \mathcal{T}^i$ be the *response space*. An LLM is a mapping from $\mathcal{X}$ to a distribution over $\mathcal{Y}$. Let $\mathcal{H}$ be the set of *human users* we are interested in. Given a context $x \in \mathcal{X}$ and two responses $(y, y') \in \mathcal{Y}^2$, we can ask a user $h \in \mathcal{H}$ for their preferred option. We use $y \succ y'|x, h$ to indicate that $h$ prefers $y$ over $y'$ given $x$ [47, 2]. We encode this outcome with the value $z = 1$, and encode the opposite preference as $z = 0$. We want to model humans' preferences as accurately as possible.

Define $\mathcal{Z} := \{0, 1\}$ and let $\mathcal{D}$ be a distribution over $\mathcal{H} \times \mathcal{X} \times \mathcal{Y}^2 \times \mathcal{Z}$. We will use subscripts to refer to the marginal distributions of $\mathcal{D}$ and superscripts to indicate conditioning. For example, $\mathcal{D}_{\mathcal{X}}^h$ is the marginal distribution over $\mathcal{X}$ conditioned on $H = h$. We will use numeric superscripts to indicate the joint distribution induced by independent draws from the same distribution; for instance, $(\mathcal{D}_{\mathcal{X}}^h)^n$ is the distribution over $\mathcal{X}^n$ resulting from $n$ independent draws from $\mathcal{D}_{\mathcal{X}}^h$.

We will formalise the problem through the lens of *empirical risk minimisation* [51]. Let $\mathcal{C}_0 := \{\mathcal{X} \times \mathcal{Y}^2 \to [0, 1]\}$ be a *classifier set* (or a *hypothesis class*) and let $\ell : [0, 1] \times \mathcal{Z} \to \mathbb{R}$ be an appropriately defined *loss function*. Our goal is to minimise the *generalisation loss* defined as $L_{\mathcal{D}}(c) := \mathbb{E}_{H,X,Y,Y',Z \sim \mathcal{D}}[\ell(c(X, Y, Y'), Z)]$. To do so, we will use data generated as follows. First, a user is sampled, $h \sim \mathcal{D}_{\mathcal{H}}$. Second, $n$ contexts are sampled, $(x_i)_{i=1}^n \sim (\mathcal{D}_{\mathcal{X}}^h)^n$. Third, for each context $x_i$, two responses are sampled, $y_i, y_i' \sim (\mathcal{D}_{\mathcal{Y}}^{h,x_i})^2$ . We then show each of the $n$ resulting tuples $(x_i, y_i, y_i')$ to $h$, who ranks them by sampling $z \sim \mathcal{D}_{\mathcal{Z}}^{h,x_i,y_i,y_i'}$ (we will refer to the users $h$ who ranked the examples as "raters"). Note that $\mathcal{D}_{\mathcal{Z}}^{h,x_i,y_i,y_i'}$ is a Bernoulli distribution whose mean is

$\mathbb{P}(y_i \succ y_i' \mid x_i, h)$, the probability that $h$ will prefer $y_i$ over $y_i'$ given $x_i$. This entire process is repeated $m$ times, resulting in the dataset $S_0 := \{(x_i, y_i, y_i', z_i)\}_{i=1}^{mn}$. We then use $S_0$ to define the *training loss*: $L_{S_0}(c) := 1/mn \sum_{i=1}^{mn} \ell(c(x_i, y_i, y_i'), z_i)$, which we will minimise as a proxy for $L_{\mathcal{D}}$ [51].

**The Bradley-Terry model.** The ultimate objective of the process above is to derive a reward function $r : \mathcal{X} \times \mathcal{Y} \to \mathbb{R}$ to be used in RL. This is possible if we assume that humans' preferences are generated based on $r$. A common assumption is the *Bradley-Terry model*: $\mathbb{P}(y \succ y' \mid x) = \sigma(r(x, y) - r(x, y'))$, where $\sigma$ is the sigmoid function [10]. Let $r_{\boldsymbol{\theta}} : \mathcal{X} \times \mathcal{Y} \to \mathbb{R}$ be a reward function parameterised by $\boldsymbol{\theta} \in \mathbb{R}^e$. This induces a set of classifiers $\mathcal{C}_0$ whose elements are $c_{\boldsymbol{\theta}}(x, y, y') := \sigma(r_{\boldsymbol{\theta}}(x, y) - r_{\boldsymbol{\theta}}(x, y'))$. We can write the likelihood of $\boldsymbol{\theta}$ with respect to $z$ as $\mathcal{L}(\boldsymbol{\theta}) = \mathbb{P}(z \mid x, y, y'; \boldsymbol{\theta}) = \sigma(r_{\boldsymbol{\theta}}(x, y) - r_{\boldsymbol{\theta}}(x, y'))^z \sigma(r_{\boldsymbol{\theta}}(x, y') - r_{\boldsymbol{\theta}}(x, y))^{1-z}$. Since the samples in the dataset $S_0$ are i.i.d., the likelihood function of $\boldsymbol{\theta}$ given the data can be written as $\mathcal{L}(\boldsymbol{\theta} \mid S_0) = \prod_i \mathbb{P}(z_i \mid x_i, y_i, y_i'; \boldsymbol{\theta})$. Thus, if we define

$$\ell(c_{\boldsymbol{\theta}}(x, y, y'), z) := -z \log c_{\boldsymbol{\theta}}(x, y, y') + (z - 1) \log c_{\boldsymbol{\theta}}(x, y', y), \tag{1}$$

we naturally have $L_{S_0}(c_{\boldsymbol{\theta}}) = -\log \mathcal{L}(\boldsymbol{\theta}|S_0)$ and $L_{\mathcal{D}}(c_{\boldsymbol{\theta}}) = -\log \mathcal{L}(\boldsymbol{\theta})$. The minimisation of $L_{S_0}(c_{\boldsymbol{\theta}})$ yields a vector of parameters $\tilde{\boldsymbol{\theta}}^* \in \mathbb{R}^e$. The corresponding $r_{\tilde{\boldsymbol{\theta}}^*}$ can then play the role of the reward function in RL, resulting in an LLM that reflects the preferences of the population $\mathcal{H}$ [15, 56].

## 3 Reward-model personalisation

One of the key assumptions underlying (1) is that

$$\mathbb{P}(y \succ y' \mid x) = \mathbb{E}_{H \sim \mathcal{D}_{\mathcal{H}}} [\mathbb{P}(y \succ y' \mid x, H)]. \tag{2}$$

That is, we are modelling the probability of response $y$ being preferred over response $y'$ given context $x$ as the average opinion among users $\mathcal{H}$ computed according to $\mathcal{D}_{\mathcal{H}}$. Clearly, (2) will only be a good assumption if the distribution $\mathcal{D}_{\mathcal{H}}$ is representative of the target users, which may or may not be the case in practice. We call attention to another issue: when we use (1) to derive a reward function, the accompanying assumption (2) implies that we are neglecting individual differences in the preferences of the population $\mathcal{H}$. Formally, we are ignoring the variance of the variable $\bar{z}(H) := p(y \succ y' \mid x, H)$.

**Preference modelling as a game.** Let us focus on the simple problem of choosing which of two possible responses $y$ and $y'$ should follow a context $x$. An intuitive way to understand the deleterious effect of making this decision based on (2) is to think of it as a game. At each round, the player picks either $y$ or $y'$. Then, a user $h \sim \mathcal{D}_{\mathcal{H}}$ is sampled and a reward is drawn from $\mathcal{D}_{\mathcal{Z}}^{x,y,y',h}$ if $y$ was selected, and from $\mathcal{D}_{\mathcal{Z}}^{x,y',y,h}$ otherwise. Clearly, the best one can do in this game is to always pick $y$ if $\mathbb{E}\bar{z}(H) \geq 0.5$, and always pick $y'$ otherwise. This results in a expected reward of $\max(\mathbb{E}\bar{z}(H), 1 - \mathbb{E}\bar{z}(H))$. We are proposing to change the game in a way that we only choose between $y$ and $y'$ *after* the user $h$ has been revealed. In this case the optimal strategy is to pick $y$ if $\bar{z}(h) \geq 0.5$, and $y'$ otherwise. This increases the expected reward to $\mathbb{E} \max(\bar{z}(H), 1 - \bar{z}(H)) \geq \max(\mathbb{E}\bar{z}(H), 1 - \mathbb{E}\bar{z}(H))$.

**User-aware reward model learning.** As the game example above suggests, our strategy will be to model $\mathbb{P}(y \succ y' \mid x, h)$ instead of $\mathbb{P}(y \succ y' \mid x)$. We will use data similar to that used in conventional RLHF, with a small extra requirement: tuples in the dataset have to be labelled with (non-identifying) rater IDs. We have $S := \{\{(h_i, x_{ij}, y_{ij}, y_{ij}', z_{ij})\}_{j=1}^n\}_{i=1}^m$, where $h_i$ is effectively an index indicating which rater determined $z_{ij}$ based on $x_{ij}$, $y_{ij}$ and $y_{ij}'$, for $j = 1, 2, ..., n$. This is not a strong requirement: we are just surfacing a piece of information used to generate $S_0$.

Given $S$, we can define an augmented set of classifiers $\mathcal{C} := \{\mathcal{H} \times \mathcal{X} \times \mathcal{Y}^2 \to \mathbb{R}\}$ which are now also a function of users $h \in \mathcal{H}$. Let $\hat{\mathcal{H}} \subseteq \mathcal{H}$ be the set of human raters $\hat{h}$ who provided feedback for the generation of $S$. We distinguish between two types of generalisation. In *intra-user generalisation* one is interested in minimising $\mathbb{E}_{\mathcal{D}}[\ell(c(H, X, Y, Y'), Z)|H \in \hat{\mathcal{H}}]$. That is, although we are able to predict preferences over contexts $x$ and responses $y$ beyond the training data, we are unable to extrapolate to users not in the set of raters $\hat{\mathcal{H}}$. In *inter-user generalisation* one is interested in minimising $L_{\mathcal{D}} := \mathbb{E}_{\mathcal{D}}[\ell(c(H, X, Y, Y'), Z)]$. That is, we are able to predict the preferences of *any user* in $\mathcal{H}$ over the entire set $\mathcal{X} \times \mathcal{Y}^2$. Our focus is on inter-user generalisation.

The current practice of not distinguishing between users accomplishes neither intra- nor inter-user generalisation, for the expectation in $L_{\mathcal{D}}$ runs over $\mathcal{H}$ but the loss $\ell$ itself is agnostic to users. This is akin to postulating a single user whose preferences coincide with the average preference in $\mathcal{H}$ (*cf.* 2).

**Theoretical analysis.**    As discussed, our goal is to minimise $L_{\mathcal{D}}$, but we only have access to $L_S$. We now analyse how well the latter tracks the former. Specifically, we will show how the difference $|L_{\mathcal{D}}(c) - L_S(c)|$ behaves with respect to the defining characteristics of the learning problem.

This is a non-standard learning theory analysis because the data in $S$ is not i.i.d. First, we need to guarantee that the loss is bounded, so we replace the $\log(\cdot)$ appearing in (1) with a counterpart defined as $\hat{\log}(\cdot) := \max(\log(\cdot), \ell_{\min})/\ell_{\min}$, where $\ell_{min}$ is large enough in magnitude to render the capping of the logarithm inconsequential; this approach is related to the truncation sometimes used in the learning theory of classification [1]. With this change (1) is guaranteed to lie in $[0, 1]$. We then define $\mathbb{V}(\ell_c) := \mathbb{V}(\ell(c(H, X, Y, Y'), Z))$, with $(H, X, Y, Y', Z) \sim \mathcal{D}$, where $\mathbb{V}(\cdot)$ stands for variance.

**Proposition 1.** *For any $c \in \mathcal{C}$, $m > 0$, $n > 0$, and $\delta \in (0, 1]$, we have with probability at least $1 - \delta$,*

$$|L_{\mathcal{D}}(c) - L_S(c)| \le \frac{1}{3m}\left[g + \sqrt{g^2 + 18gm\left(\frac{1}{n}\mathbb{E}[\mathbb{V}(\ell_c|H)] + \mathbb{V}(\mathbb{E}[\ell_c|H])\right)}\right], \qquad (3)$$

*where $g := \ln(2/\delta)$.*

The proofs of our theoretical results are in Appendix A. We will refer to the bound on the right-hand side of (3) as $\epsilon$. In line with similar results in the literature, $\epsilon$ is $O(\ln(1/\delta))$ as $\delta \to 0^+$ [51]. Perhaps more surprising is the fact that, as the number of examples per rater $n$ tends to $\infty$, $\epsilon$ approaches a constant. On closer inspection this makes intuitive sense: with a fixed number of raters $m$, there is an irreducible error that cannot be fully eliminated even if we collect an infinite number of examples $n$ per rater. In contrast, $\epsilon \to 0$ as $m \to \infty$, regardless of $n$, at a rate of $O(1/\sqrt{m})$. This is consistent with the literature, and also makes sense, as we can sample the same rater $h$ over and over (and thus even with a single $n = 1$ example per rater we will eventually have a perfect estimation).

Another interesting observation for the discussion herein is the dependency of $\epsilon$ on the terms $\mathbb{E}[\mathbb{V}(\ell_c|H)]$ and $\mathbb{V}(\mathbb{E}[\ell_c|H])$. Intuitively, $\mathbb{E}[\mathbb{V}(\ell_c|H)]$ measures how much the loss varies per user, on average, while $\mathbb{V}(\mathbb{E}[\ell_c|H])$ measures how much the average loss varies across users. If we keep all other variables fixed in (3), we have that $\epsilon \propto \sqrt{\mathbb{E}[\mathbb{V}(\ell_c|H)] + \mathbb{V}(\mathbb{E}[\ell_c|H])}$. Clearly, we want to make both terms as small as possible. This sheds light on the discussion above: while increasing the number of examples per rater $n$ weakens the effect of $\mathbb{E}[\mathbb{V}(\ell_c|H)]$ on the estimate $L_S(c)$, it does not change the effect of $\mathbb{V}(\mathbb{E}[\ell_c|H])$, which can only be mitigated with a larger number of raters $m$.

The discussion above invites the question of how to best allocate a budget of $k = nm$ training examples. The right-hand side of our bound (3) is minimised when $n = 1$ and $m = k$, which suggests we should sample as many raters from $\mathcal{H}$ as training examples (with replacement). While it may not always be feasible to have many raters providing only a few examples each, understanding the trade-offs elicited by our theoretical results may guide the allocation of resources in practice.

The bound in (3) shows how the empirical loss $L_S$ deviates from the true generalisation loss $L_{\mathcal{D}}$ for a *single* classifier $c \in \mathcal{C}$. Since we are usually interested in finding a classifier that minimises $L_{\mathcal{D}}$, ideally we would be able to bound $|L_{\mathcal{D}}(c) - L_S(c)|$ over the entire set $\mathcal{C}$. This is what we set out to do next. Let $c^* := \arg\min_{c \in \mathcal{C}} L_{\mathcal{D}}(c)$ and let $\tilde{c}^* := \arg\min_{c \in \mathcal{C}} L_S(c)$. That is, $c^*$ is the true minimiser of $L_{\mathcal{D}}$ in $\mathcal{C}$ and $\tilde{c}^*$ is the classifier we obtain by minimising the empirical loss $L_S$. We will bound the difference $L_{\mathcal{D}}(\tilde{c}^*) - L_{\mathcal{D}}(c^*)$.

First define $\ell_{\mathcal{C}} := \ell_{\arg\sup_{c \in \mathcal{C}} \mathbb{V}(\ell_c)}$. We will also need the concept of a covering number for $\mathcal{C}$. Given $\alpha \in [0, 1]$, let $\mathcal{C}_{\alpha} \subseteq \mathcal{C}$ be the smallest set of classifiers such that

$$\forall_{c \in \mathcal{C}} \exists_{c' \in \mathcal{C}_{\alpha}} \forall_{h \in \mathcal{H}, x \in \mathcal{X}, y \in \mathcal{Y}, y' \in \mathcal{Y}', z \in \mathcal{Z}} |\ell(c(h, x, y, y'), z) - \ell(c'(h, x, y, y'), z)| \le \alpha.$$

In words, $\mathcal{C}_{\alpha}$ is the smallest set that *covers* $\mathcal{C}$ with respect to $\ell$: for every $c \in \mathcal{C}$, there is at least one $c' \in \mathcal{C}_{\alpha}$ whose loss is within a distance of $\alpha$ from that of $c$. We use $|\mathcal{C}_{\alpha}|$ as a measure of the complexity or "capacity" of $\mathcal{C}$. It follows from the definition that there exists $\alpha \in [0, 1]$ such that $|\mathcal{C}_{\alpha}| < \infty$ (for example, $|\mathcal{C}_{\alpha=1}| = 1$).

**Proposition 2.** *For any $m, n > 0$ and any $\delta \in (0, 1]$, we have with probability at least $1 - \delta$ that*

$$L_{\mathcal{D}}(\tilde{c}^*) \le L_{\mathcal{D}}(c^*) + 2\inf_{\alpha}\left[\frac{1}{3m}\left(g_{\alpha} + \sqrt{g_{\alpha}^2 + 18mg_{\alpha}\left(\frac{1}{n}\mathbb{E}[\mathbb{V}(\ell_{\mathcal{C}}|H)] + \mathbb{V}(\mathbb{E}[\ell_{\mathcal{C}}|H])\right)}\right) + 2\alpha\right],$$
$$(4)$$

*where $g_{\alpha} := \ln(2|\mathcal{C}_{\alpha}|/\delta)$.*

Expression (4) is similar to (3), the main difference being its dependency on $|\mathcal{C}_\alpha|$, our measure of the complexity of $\mathcal{C}$. Fortunately the bound grows with $\ln(|\mathcal{C}_\alpha|)$, which means that we can afford to substantially increase the "size" of $\mathcal{C}$ without incurring too high a cost in terms of the number $m$ of raters and the number $n$ of training examples per rater needed. The precise value of $|\mathcal{C}_\alpha|$ will depend on $\alpha$ and the set $\mathcal{C}$. We can show that $|\mathcal{C}_\alpha| \leq (1/\alpha)^{|\mathcal{H}||\mathcal{X}||\mathcal{Y}|^2}$ (see Proposition 3 in Appendix A.2). It is possible to derive bounds tighter than (4) if one is willing to make assumptions about $\mathcal{C}$, use measures of complexity other than $|\mathcal{C}_\alpha|$, or both. See Proposition 4 in Appendix A.2 for an example in which combining logistic regression models and the notion of Rademacher complexity leads to improved variance dependence.

The best possible generalisation error $L_\mathcal{D}(c^*)$ will depend on two factors: how well the classifiers in $\mathcal{C}$ can model different users and how heterogenous the population $\mathcal{H}$ is in terms of preferences. Consider a set $\mathcal{C}_0$ whose classifiers cannot distinguish between users. We will construct a more general set $\mathcal{C}$ of classifiers that can differentiate between users by imposing that, for any $c \in \mathcal{C}$, $c(h, \cdot) \in \mathcal{C}_0$ (in words, this means that any classifier in $\mathcal{C}$ reduces to a classifier in $\mathcal{C}_0$ if we fix the user $h$). We will also make sure that all classifiers in $\mathcal{C}_0$ also belong to $\mathcal{C}$, so we have $\mathcal{C}_0 \subseteq \mathcal{C}$. It follows from the latter assumption that, by definition, $|\mathcal{C}_\alpha| \geq |(\mathcal{C}_0)_\alpha|$ and $\mathbb{V}(\ell_\mathcal{C}) \geq \mathbb{V}(\ell_{\mathcal{C}_0})$.

Let us now analyse two scenarios. In the extreme scenario where all users in $\mathcal{H}$ have the same preference, it is clear that the best classifier in $\mathcal{C}$ also belongs to $\mathcal{C}_0$, so considering the former instead of the latter will not decrease $L_\mathcal{D}(c^*)$. Using the fact that in this scenario $\mathbb{V}(\mathbb{E}[\ell_{\mathcal{C}_0}|H]) = 0$, it is easy to show that replacing $\mathcal{C}_0$ with $\mathcal{C}$ cannot decrease, and will in general increase, the right-hand side of (4). That is, we may need more training examples to achieve the same generalisation error. This formalises the intuition that, when users agree with each other, having a smaller classifier set $\mathcal{C}_0$ whose classifiers are agnostic to users is often advantageous.

Next we turn our attention to the more realistic scenario where users do disagree with each other. In this case considering $\mathcal{C}$ instead of $\mathcal{C}_0$ can yield a lower optimal population loss $L_\mathcal{D}(c^*)$, as illustrated by the game example following Equation (2). Interestingly, because in this scenario $\mathbb{V}(\mathbb{E}[\ell_{\mathcal{C}_0}|H])$ is not necessarily zero, and in fact can be quite large, it may be the case that $\mathbb{V}(\mathbb{E}[\ell_\mathcal{C}|H]) < \mathbb{V}(\mathbb{E}[\ell_{\mathcal{C}_0}|H])$ (which implies that $\mathbb{E}[\mathbb{V}(\ell_\mathcal{C}|H)] > \mathbb{E}[\mathbb{V}(\ell_{\mathcal{C}_0}|H)]$, as $\mathbb{V}(\ell_\mathcal{C}) \geq \mathbb{V}(\ell_{\mathcal{C}_0})$). This means that, for large enough $n$, replacing $\mathcal{C}_0$ with $\mathcal{C}$ can in fact *decrease* the effect of the loss variance on the bound (4). The overall bound will not necessarily get smaller, though, as $|\mathcal{C}_\alpha|$ will show up instead of $|(\mathcal{C}_0)_\alpha|$. Regardless of its effect on the bound (or, more generally, on the resulting sample complexity), adopting a set $\mathcal{C}$ of classifiers that distinguish between users may be highly beneficial when users disagree with each other, as the decrease in the optimal generalisation error $L_\mathcal{D}(c^*)$ can be substantial.

## 4    Reward-feature models

The theoretical results presented in the previous section apply to any classifier set $\mathcal{C}$; we now discuss specific ways of defining this set and argue for one architecture in particular.

As discussed in Section 2, the classifiers $c_{\boldsymbol{\theta}}$ function like a "wrapper" around the reward functions $r_{\boldsymbol{\theta}}$, which are the objects we are really interested in. We advocate the use of reward models with two disjoint sets of parameters: a common vector of parameters $\boldsymbol{\theta} \in \mathbb{R}^e$ that is shared among all users $h \in \mathcal{H}$, including those $h \notin \hat{\mathcal{H}}$, and parameters $\boldsymbol{\theta}_h \in \mathbb{R}^d$ that are specific to user $h$ (this is closely related to the concept of "adapters" [50, 30, 45]). We will call $r_{\boldsymbol{\theta}, \boldsymbol{\theta}_h}$ *adaptive reward models*.

The partitioning of the parameters of adaptive reward models induces a division of their training procedure itself. The data in $S$ can be used to learn $\boldsymbol{\theta}$ and $\boldsymbol{\theta}_{\hat{h}}$ for a rater $\hat{h} \in \hat{\mathcal{H}}$. However, to learn the parameters $\boldsymbol{\theta}_h$ associated with a user not in $S$, $h \in \mathcal{H} \setminus \hat{\mathcal{H}}$, one needs additional data containing feedback provided by $h$, $S_h := \{(x_i, y_i, y_i', z_i)\}_{i=1}^{\hat{n}}$, with $z_i \sim \mathcal{D}_\mathcal{Z}^{x_i, y_i, y_i', h}$. We will refer to the use of $S$ to learn $\boldsymbol{\theta}$ and $\boldsymbol{\theta}_{\hat{h}}$ as *training*; learning $\boldsymbol{\theta}_h$ using $S_h$ will be referred to as *adaptation*.

We assume we have abundant data for training in $S$, but adaptation must take place with as few additional training examples as possible in $S_h$ (that is, $n \gg \hat{n}$). Because of this, and also because we want the adaptation of the model to a specific user $h \in \mathcal{H} \setminus \hat{\mathcal{H}}$ to be quick, we generally want to have $e \gg d$—in words, we want the number of shared parameters to be much larger than the number of parameters that are specific to a given individual. Another, slightly more practical reason to have a large number of shared parameters is that this allows for a distributed learning architecture in which the learning of $\boldsymbol{\theta}$ can make use of a centralised, powerful computational infrastructure, while each $\boldsymbol{\theta}_h$ can be learned "locally" using less resources.

**Reward features.** We now discuss an architecture for $r_{\boldsymbol{\theta}, \boldsymbol{\theta}_h}$ that has particularly nice properties. We define *reward features* as a function $\boldsymbol{\phi}(x, y) : \mathcal{X} \times \mathcal{Y} \to \mathbb{R}^d$. We then assume that the reward for individual $h$ is given by

$$r_h(x, y) = \langle \boldsymbol{\phi}(x, y), \mathbf{w}_h \rangle, \tag{5}$$

where $\mathbf{w}_h \in \mathbb{R}^d$ and $\langle \cdot, \cdot \rangle$ denotes inner product. Following the steps taken in Section 2, we define a parametric form for $r_h$ that reflects (5). We first define a parameterised function $\boldsymbol{\phi}_{\boldsymbol{\theta}} : \mathcal{X} \times \mathcal{Y} \to \mathbb{R}^d$, with $\boldsymbol{\theta} \in \mathbb{R}^e$. Note that $\boldsymbol{\theta}$ is the set of shared parameters defined above; the parameters $\boldsymbol{\theta}_h$ associated with a specific $h \in \mathcal{H}$ are simply a vector $\mathbf{w}_h \in \mathbb{R}^d$. Putting it all together, our parameterised model is $r_{\boldsymbol{\theta}, \mathbf{w}_h}(x, y) = \langle \boldsymbol{\phi}_{\boldsymbol{\theta}}(x, y), \mathbf{w}_h \rangle$. We call this architecture a *reward-feature model* (RFM).

**Training.** We now derive an appropriate optimisation objective for RFM's training. Let $\mathbf{W} \in \mathbb{R}^{|\hat{\mathcal{H}}| \times d}$ be a matrix formed by stacking $|\hat{\mathcal{H}}|$ vectors $\mathbf{w}_{\hat{h}}$. Let $c_{\boldsymbol{\theta}, \mathbf{W}}(\hat{h}, x, y, y') \coloneqq \langle \boldsymbol{\phi}_{\boldsymbol{\theta}}(x, y) - \boldsymbol{\phi}_{\boldsymbol{\theta}}(x, y'), \mathbf{w}_{\hat{h}} \rangle$, where $\mathbf{w}_{\hat{h}}$ is the row of $\mathbf{W}$ corresponding to $\hat{h}$. Plugging $c_{\boldsymbol{\theta}, \mathbf{W}}$ into (1), we get

$$\ell(c_{\boldsymbol{\theta}, \mathbf{W}}(\hat{h}, x, y, y'), z) \coloneqq -z \log\langle \boldsymbol{\phi}_{\boldsymbol{\theta}}(x, y) - \boldsymbol{\phi}_{\boldsymbol{\theta}}(x, y'), \mathbf{w}_{\hat{h}} \rangle + (z - 1) \log\langle \boldsymbol{\phi}_{\boldsymbol{\theta}}(x, y') - \boldsymbol{\phi}_{\boldsymbol{\theta}}(x, y), \mathbf{w}_{\hat{h}} \rangle. \tag{6}$$

As before, if we minimise the empirical loss $L_S$ induced by (6), we will automatically be maximising the likelihood $\mathcal{L}(\boldsymbol{\theta}, \mathbf{W}|S)$, which is a proxy for $\mathcal{L}(\boldsymbol{\theta}, \mathbf{W})$.

RFM's training yields a feature function $\boldsymbol{\phi}_{\boldsymbol{\theta}} : \mathcal{X} \times \mathcal{Y} \to \mathbb{R}^d$ and a matrix $\mathbf{W} \in \mathbb{R}^{|\hat{\mathcal{H}}| \times d}$. Each dimension of $\boldsymbol{\phi}_{\boldsymbol{\theta}}$, $(\boldsymbol{\phi}_{\boldsymbol{\theta}})_i$, can be interpreted as a criterion used by humans to express their preferences. The $i$-th element of $\mathbf{w}_{\hat{h}}$, $w_{\hat{h}i}$, represents how much rater $\hat{h}$ values (or does not value) criterion $(\boldsymbol{\phi}_{\boldsymbol{\theta}})_i$. Importantly, learning $\boldsymbol{\phi}_{\boldsymbol{\theta}}$ does not depend on raters being able to articulate the criteria underlying their preferences; the only thing that is required from them is to rank pairs of candidate responses.

The question arises as to how well the features $\boldsymbol{\phi}_{\boldsymbol{\theta}}$ will be able to capture the preferences of the entire population $\mathcal{H}$. We need $\boldsymbol{\phi}_{\boldsymbol{\theta}}$ to have enough representational capacity—a proxy for which is the number of parameters $e$—and its dimension $d$ to be sufficient to capture the preferences in $S$. However, when $e$ and $d$ are too large, an RFM may effectively become $|\hat{\mathcal{H}}|$ independent reward models. We usually do not want that. Instead, we want $\boldsymbol{\phi}_{\boldsymbol{\theta}}$ to capture features that are *common* to the raters in $\hat{\mathcal{H}}$, since these should also reflect the preferences of the users in $\mathcal{H}$ more generally. This means that $e$ and $d$ have to be appropriately set or controlled through regularisation.

**Adaptation.** We are interested in using features $\boldsymbol{\phi}_{\boldsymbol{\theta}}$ learned during training to specialise $r_{\boldsymbol{\theta}, \mathbf{w}_h}$ to users beyond the raters $\hat{h} \in \hat{\mathcal{H}}$. The problem of adapting $r_{\boldsymbol{\theta}, \mathbf{w}_h}$ to an unseen user can be formulated as a small modification of training in which the loss (6) is minimised with respect to a new set of coefficients $\mathbf{w}$ while the features $\boldsymbol{\phi}_{\boldsymbol{\theta}}$ are held fixed. Formally, we define a classifier set parameterised by $\mathbf{w}$, $c_{\mathbf{w}}(h, x, y, y') \coloneqq \langle \boldsymbol{\phi}_{\boldsymbol{\theta}}(x, y) - \boldsymbol{\phi}_{\boldsymbol{\theta}}(x, y'), \mathbf{w} \rangle$, plug it into (6), and minimise $L_{S_h}$ with respect to $\mathbf{w}$. Adaptation is particularly simple with an RFM: since $\boldsymbol{\theta}$ is frozen, optimising $\mathbf{w}$ comes down to logistic regression, which is a well understood convex optimization problem whose sample complexity has been characterised for different scenarios (see, for example, [37, 43, 31] and references therein).

**Connection with the theory.** We now discuss how to connect RFM's two-stage learning process with the theory developed in Section 3. First note that the upper bound in (4) directly applies to RFM's intra-user generalisation loss, for in this case adaptation is not necessary. To apply a version of the bound (4) to training *and* adaptation, we must see RFM as a function of $\boldsymbol{\theta}$ only. We can accomplish this by folding adaptation into the functioning of the model. When faced with a new tuple $(\hat{h}, x, y, y')$, we create a dataset $S_h$ on-the-fly by sampling $\hat{n}$ examples from $\mathcal{D}^h$, and then make $\mathbf{w}_h = \arg\min_{\mathbf{w}} L_{S_h}(c_{\mathbf{w}})$. That is, $\mathbf{w}_h$ is no longer a free parameter of the model. If we use the model in this way during training, we can use (4) to bound its post-adaption generalisation error.

If we are concerned with adaptation only, (4) directly applies as a special case when $|\mathcal{H}| = 1$, which greatly simplifies the bound but obscures some of the underlying insights. Because RFM's adaptation is a convex optimisation problem, there are known generalisation bounds that scale nicely, avoiding polynomial dependence on factors such as $|\mathcal{X}|$ and $|\mathcal{Y}|$ (see for example [1, Theorem 6.1]).

**Why not other adaptive reward model architectures?** As discussed, RFMs are but one instantiation of a more general class of adaptive reward models $r_{\boldsymbol{\theta}, \boldsymbol{\theta}_h}$. We now lay out a few arguments supporting our choice of the RFM architecture in particular.

First, as discussed, the fact that RFM's architecture is linear in $\mathbf{w}_h$ gives rise to a convex adaptation problem. This should not be overlooked: having a model able to reliably adapt to new users in a low-data regime may be crucial [51]. Second, and relatedly, RFM's simple architecture also makes it easier to improve the entire training pipeline, as one can resort to well established linear algebra techniques [52]. Third, the fact that the model's output is a linear combination of the features $\phi_{\boldsymbol{\theta}}$ should make it easier to interpret their contributions (note that $\phi_{\boldsymbol{\theta}}$ itself can be an arbitrarily complex non-linear functions of the inputs) [38]. Fourth, it should be easy to add new, possibly handcrafted, features that represent agreed upon metrics like safety, helpfulness, and factuality [3, 57, 21].

But perhaps the main reason to adopt RFMs is the fact that they allow for an easy adaptation of the downstream LLM (or, more generally, policy). Recall that ultimately we are interested in using $r_{\boldsymbol{\theta}, \mathbf{w}_h}$ to steer the behaviour of a policy. There are methods in the literature specifically designed to synthesise a policy that performs well under a linear combination of features whose coefficients are only provided at deployment time [4–6, 29, 32, 7, 33, 49, 53, 14, 11]. In our case, this means that, after adapting to a user using a few examples, we can hand the resulting $\mathbf{w}_h$ over to one of these methods and immediately obtain a policy (or LLM) that is specialised to the corresponding reward.

**Limitations.** There is an inherent tension between training and adaptation: in general, the simpler the latter, the more demanding the former becomes. RFM sits on one extreme of this spectrum, with an adaptation as simple as it can be at the expense of a potentially complex training. While this aligns with current LLM practices—with ample data and compute for offline training and scarcer resources for adaptation—, it might not be the ideal trade-off in other contexts, or even for LLMs if resource availability shifts in the future.

## 5 Experiments

We now present experiments illustrating our findings. Due to space constraints we only describe the crucial aspects of our experimental setup; for further details and additional results, see Appendix B.

We adopted UltraFeedback, a dataset carefully curated to ensure the quality and diversity of the responses [18]. This is a relatively large dataset for this type of study: the version we adopted has a training set with $60,829$ examples and a test set with $985$ examples.[1] On the solution side, we used Google DeepMind's [26] Gemma 1.1 2B model to implement both a baseline and RFM.[2] The baseline is a reward model that neither distinguishes between raters nor performs adaptation, as is common practice. We implemented RFM using the same Gemma model, with the final layer replaced by $d$ counterparts corresponding to the features $\phi_{\boldsymbol{\theta}}$. Both models were trained using gradient descent to minimise their corresponding losses, starting from the pre-trained parameters of Gemma.

The goal of our experiments is twofold: to illustrate the theoretical results in Section 3 and to assess RFM's performance as compared to existing baselines. To study both rigorously, we must be able to vary the number of raters as well as their heterogeneity in terms of preferences. We accomplished this by defining a systematic way of generating synthetic raters with different characteristics. We created 13 features $\{\phi_i(x, y)\}_{i=1}^{13}$ capturing different aspects of a context-response pair: from superficial, easy-to-compute features, like length or the number of adverbs in $y$, to more semantic features such as the number of words in $y$ that are synonyms (or antonyms) of words in $x$ (see Appendix B.1 for details). In contrast with features generated by LLMs, even our most nuanced features can be unequivocally and efficiently computed. They are also devoid of inherent valence.

We defined a user as a vector $\boldsymbol{\omega} \in \{-1, 1\}^{13}$, that is, a user either likes ($+1$) or dislikes ($-1$) each one of the features $\phi_i$. This gives rise to a space $\mathcal{H}$ with $2^{13}$ users. Given an example $(x, y, y')$ and a user $\boldsymbol{\omega}$, we determine the corresponding preference as $z = \mathbb{1}\{\langle \phi(x, y), \boldsymbol{\omega} \rangle > \langle \phi(x, y'), \boldsymbol{\omega} \rangle\}$, where $\mathbb{1}\{\cdot\}$ is the indicator function. We defined a distribution $\mathcal{D}_{\mathcal{H}}$ parameterised by a single scalar $p \in [0, 1]$ indicating the probability that each $\phi_i$ is liked independently of the others. That is, we sample users $h \sim \mathcal{D}_{\mathcal{H}}$ by drawing each $\omega_{hi}$ from a Bernoulli with mean $p$. For each experiment, we sampled $m$ raters to rank the training set and 500 held-out users to assess the inter-user generalisation of the models on the test set (this number is considerably larger than in previous studies [44, 13, 52]).

It is worth emphasising that throughout our experiments RFM did *not* have access to the real features $\phi$ underlying the data. Instead, it *learned* features $\phi_{\boldsymbol{\theta}}$ parameterised by $\boldsymbol{\theta} \in \mathbb{R}^e$ using (6). The notation RFM($d$) indicates that a $d$-dimensional $\phi_{\boldsymbol{\theta}} \in \mathbb{R}^d$ was learned.

---

[1] huggingface.co/datasets/allenai/ultrafeedback_binarized_cleaned

[2] huggingface.co/google/gemma-1.1-2b-it

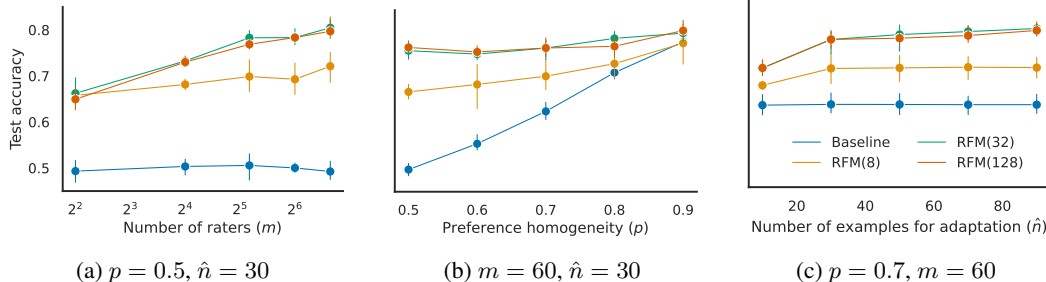

(a) $p = 0.5$, $\hat{n} = 30$  (b) $m = 60$, $\hat{n} = 30$  (c) $p = 0.7$, $m = 60$

Figure 1: Accuracy in predicting the preferences of 500 held-out users on the test set after adaptation (estimate of inter-user generalisation). Error bars are 99% confidence intervals over 5 runs.

**Empirical analysis.** Figure 1 shows the performances of the baseline and RFM as we vary the number $m$ of training raters, the parameter $p$ underlying $\mathcal{D}_{\mathcal{H}}$, the number $\hat{n}$ of adaptation examples, and the number $d$ of features learned by RFM. Note how RFM behaves as predicted by our theoretical results, with performance improving with the number of raters $m$. Also as predicted by the theory, the performance of both the baseline and RFM improve with the homogeneity $p$ of the users' preferences ($\mathcal{D}_{\mathcal{H}}$ has maximum entropy when $p = 0.5$, so preferences get more homogeneous as $p \to 1$). However, RFM's performance is much more robust to changes in $p$, with almost no degradation when a sufficient number of features $d \geq 32$ are learned. Also note how RFM can adapt well to a new user using as few as $\hat{n} = 30$ examples, and for $d \in \{32, 128\}$ performance keeps increasing with $\hat{n}$ (albeit only slightly). This means that, by ranking only 30 examples, a user can replace a generic reward model reflecting the average opinion of the population $\mathcal{H}$ with a model specialised to them.

**Modulating the LLM's output.** Next we assess how well RFM's good performance transfers to the scenario where it is used to steer the behaviour of an LLM. We use *best-of-n* over up to $n = 40$ responses to each context $x$ in UltraFeedback's test set [55]. The responses were generated by Google DeepMind's [27] Gemma 2 9B and Gemma 2 27B (20 responses each). For each context $x$ in the test set, we scored all 40 responses using the baseline and RFM(32) adapted with $\hat{n} = 30$ examples. Next, we selected the best-of-$n$ response to context $x$ according to the baseline and RFM, and declared either a winning model or a draw based on the actual score of their selected responses computed by user $\boldsymbol{\omega}$. Figure 2 shows results when best-of-$n$ is applied with increasing $n$. We highlight that the 40 candidate responses used come from a distribution that is different from the one used for training. Yet, RFM consistently outperforms the baseline, and the fraction of times its selected response is preferred grows with $n$.

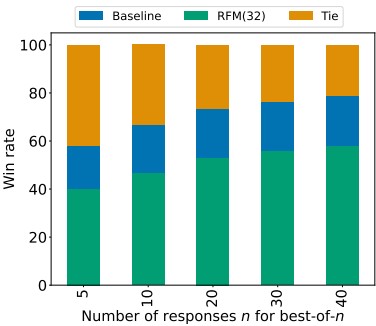

Figure 2: Relative accuracy in predicting the preferences of 500 held-out users using $\hat{n} = 30$ examples.

**Comparisons.** We now compare RFM with three types of adaptive counterparts. The first one is a linear model that adapts to a user $h$ by fine-tuning the final layer of the (trained) baseline through gradient descent on the corresponding dataset $S_h$. This is similar to RFM, except that the baseline features being linearly combined were learned without raters $h$ being distinguished (and thus this is a clear way to assess whether doing so is indeed beneficial). The second comparison involves the non-linear architecture proposed by Park et al. [42]. We partitioned the shared parameters $\boldsymbol{\theta} = [\boldsymbol{\theta}_1, \boldsymbol{\theta}_2]$ and learned $\boldsymbol{\theta}_2$ together with the vectors $\mathbf{w}_h$. This was implemented by freezing the backbone Gemma model ($\boldsymbol{\theta}_1$) and representing $\phi$ as a multilayer perceptron (MLP) with 3 or 5 hidden layers containing 32 units each ($\boldsymbol{\theta}_2$; details in Appendix B.2). The third comparison is with models that perform "in-context" adaptation. We implemented these using two prominent LLMs, Google DeepMind's [28] Gemini 1.5 Pro and OpenAI's [40] GPT-4o. To assess the LLMs' prediction accuracy for held-out user $h$, we provided $\hat{n} = 10$ training examples ranked by $h$ together with the test example to be ranked (the precise prompt can be found in Appendix B.2). For reference, we also show the "zero-shot" performance of Gemini, obtained with $\hat{n} = 0$.

Figure 3 shows the results of the two LLMs side-by-side with those obtained by the baseline, the linear baseline, the non-linear baselines, and RFM(32)—the last three also adapted using $\hat{n} = 10$ examples. The linear baseline performs on par with its non-adaptive counterpart, which suggests that features learned in a user-agnostic way are not capable of capturing the variance of the population $\mathcal{H}$. The non-linear baselines perform poorly, probably because they have too many parameters to be trained with only $\hat{n} = 10$ examples. Surprisingly, the LLMs also seem to be unable to capture the users' preferences, with their performance reducing to chance (see further discussion in Appendix B.2). RFM correctly predicts the users' preferences around $70\%$ of the time.

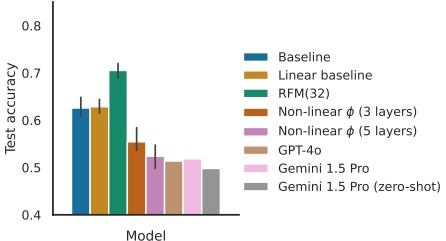

Figure 3: Accuracy in predicting the preferences of 500 held-out users using $\hat{n} = 10$ examples for adaptation. Error bars are $99\%$ confidence intervals over 5 runs.

We also compared RFM with Poddar et al.'s [44] *variatonal preference learning* (VPL), described in Section 6. Even though in principle VPL allows for inter-user generalisation, Poddar et al. have only assessed VPL's intra-user generalisation. They report an estimate of this type of generalisation on the UltraFeedback dataset, using the features provided with it, of $61.49\%$. We tried to reproduce their experimental protocol as closely as possible, as explained in Appendix B.2. The resulting estimate of RFM(32)'s intra-user generalisation is $61.61\%$, which is on par with VPL's.

**Modelling groups of real users.** In the previous experiments we had access to the features $\phi$ underlying the raters' preferences; although this is useful to investigate the models' performance under different conditions, in a more realistic scenario we would only have the training examples in the dataset $S$. To simulate this situation, in this section we use reward models as the users $h$.

We performed experiments using two datasets: UltraFeedback, as before, and also Zollo et al.'s [65] PersonalLLM. Each example in the PersonalLLM dataset has been scored by 10 reward models. To mirror this setup with UltraFeedback, we used 8 publicly-available reward models to score and rank all its test examples. We provide a detailed account of our experimental setup in Appendix B.3.

All the reward models considered were trained with human preference data, and hence they reflect the opinion of groups of real people. As these models do not distinguish between raters—much like the baseline—, they reflect an average over preferences, and thus tend to "agree" considerably with each other. To avoid such overlapping, which may render the distinction of raters unnecessary, we filtered out all the examples in the training and test sets of both datasets in which two or fewer raters disagreed with the majority. We then carried out a "leave-one-out" cross-validation composed of $k$ rounds in which $k - 1$ of the models played the role of the raters $\hat{\mathcal{H}}$ and the remaining model played the role of the held-out user ($k = 8$ for UltraFeedback and $k = 10$ for PersonalLLM).

Results are shown in Figure 4. We compare RFM with the non-adaptive and linear baselines used in the previous experiments. RFM's performance either matches or significantly surpasses that of the baselines in most cases, suggesting that it can be useful in real scenarios. As the reward models in this experiment reflect the preferences of aggregated real users, some of the heterogeneity of the population's preferences has probably been smoothed out in the training of these models. We conjecture that if the reward models were replaced by the real users underlying these models, RFM's advantage over the two baselines would be greater still.

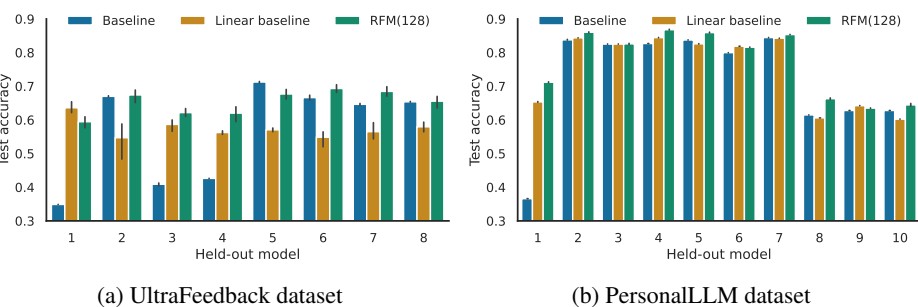

(a) UltraFeedback dataset  (b) PersonalLLM dataset

Figure 4: Accuracy in predicting the preferences of a held-out reward model using $\hat{n} = 50$ examples for adaptation. Error bars are $99\%$ confidence intervals.

# 6 Related work

We distinguish between two categories of adaptive reward models. In the first we have methods whose goal is to compute a single policy representing some form of compromise across users [22, 12, 54, 16, 19, 61]. These methods take individual idiosyncrasies into account primarily to build more accurate models, and hence they do not necessarily provide a mechanism for adapting to new users.

In the second category we have methods designed to support the downstream specialisation of a policy to a specific user, like RFMs [44, 63, 24, 36, 34, 13, 17, 25, 35, 42, 64, 52, 8]. Among these, Shenfeld et al.'s [52] and Bose et al.'s [8] works are probably the closest to ours. They both concurrently propose the same architecture as RFM, though with different names and slightly different emphases. While we analyse the *learning problem*—how training and adaptation change as the dataset $S$ changes—, Shenfeld et al. focus on how to solve the resulting *optimisation problem*: how to go about training and adaptation once we have committed to a specific dataset $S$. Shenfeld et al. also show how to leverage RFM's simple architecture to improve several aspects of the training pipeline, most notably a stable initialisation scheme via singular value decomposition of the data matrix and a more efficient way of selecting adaptation examples based on active learning. Bose et al. provide a particularly clear exposition of the subject and a well-executed empirical evaluation that includes comparisons and datasets not considered here. We see these two concurrent works as highly complementary to our own, as they collectively provide mutually reinforcing theoretical arguments and empirical evidence in favour of the proposed architecture.

Poddar et al.'s [44] *variatonal preference learning* (VPL) encodes examples previously ranked by a user into a latent variable, and then conditions the reward model and the policy on that variable (we compare VPL with RFM in Section 5). Zhao et al. [63] also encode pairs of ranked responses, but instead of resorting to variational techniques they use an LLM to do in-context inference of the rewards. Similar methods encode different types of information about users, like demographic data, either in isolation or together with examples of ranked pairs [24, 36, 34]. Chen et al. [13] propose to replace the Bradley-Terry model with Coombs's [17] *ideal point model*. The resulting method is similar to ours, albeit slightly more complex. Go et al.'s [25] approach is also similar to ours, but they use features computed by LLMs with handcrafted prompts.

We redirect the reader to Appendix C for an in-depth discussion of some of the works cited above.

# 7 Conclusion

We have formalised and analysed the problem of learning a reward model that can adapt to users. To the best of our knowledge, this is the first time this problem is rigorously studied under the assumptions considered. We have derived a PAC bound that elicits the dependency of the approximation error on the number of training examples and raters. Our analysis provides a formal framework for assessing the trade-offs involved in the collection of preference data and the use of an adaptive reward model.

We have also introduced RFM, a reward-model architecture specifically designed for fast adaptation to new users. RFM can be trained using pairwise response comparisons provided by humans. This results in a set of reward features that can be linearly combined to represent a user, even if their preferences are not reflected in the training data. Such an adaptation process can be formulated as a simple logistic regression—a well-understood convex classification problem. We showed how RFM can be personalised to an unknown user using a few dozen pairs of examples ranked by them. We have presented experiments showing how RFM compares favourably with a non-adaptive baseline, which is today's prevailing practice, and also several adaptive counterparts.

RFM can be readily combined with zero-shot RL methods that construct a policy on-the-fly which performs well under a linear combination of features. In the context of LLMs, this means that by responding to a few questions a user can have an otherwise generic model specialised to their taste. Although we have focused on the use of RFM in the context of LLMs, it is also applicable to other modalities beyond language, like images, sound, and video. In fact, RFM can be used as a reward function in any scenario that can be formalised as an RLHF problem, which includes not only more general state and action spaces but also multi-step interactions of the policy with the environment.

## Acknowledgements

We would like to thank Tom Schaul, Benjamin Van Roy, and Dan Andrei Calian for their insightful comments and useful feedback. We are also thankful to Bernardo Ávila Pires for assistance with one of the experiments and to Canfer Akbulut and Arianna Manzini for contributing to the discussion regarding the potential broader impact of our work. Finally, we would like to thank the anonymous reviewers for their comments and suggestions.

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

# A  Proofs and additional theoretical results

In this section we prove the theoretical results in the main paper and present two complementary results. To simplify the notation, we define $\mathcal{K} := \mathcal{X} \times \mathcal{Y}^2$ and use $k$ to refer to tuples $(x, y, y')$.

## A.1  Proofs of the theoretical results in the main paper

We restate the results presented in the main paper and prove them.

**Proposition 1.** *For any $c \in \mathcal{C}$, $m > 0$, $n > 0$, and $\delta \in (0, 1]$, we have with probability at least $1 - \delta$ that*

$$|L_{\mathcal{D}}(c) - L_S(c)| \le \frac{1}{3m} \left[ g + \sqrt{g^2 + 18gm \left( \frac{1}{n} \mathbb{E}[\mathbb{V}(\ell_c|H)] + \mathbb{V}(\mathbb{E}[\ell_c|H]) \right)} \right],$$

*where $g := \ln(2/\delta)$.*

*Proof.* Let $\mathbf{k}^n \in \mathcal{K}^n$ and $\mathbf{z}^n \in \mathcal{Z}^n$, and define

$$\ell_n(c, h, \mathbf{k}^n, \mathbf{z}^n) := \frac{1}{n} \sum_{i=1}^{n} \ell(c, h, k_i, z_i). \tag{7}$$

We can write

$$|L_{\mathcal{D}}(c) - L_S(c)| = \left| L_{\mathcal{D}}(c) - \frac{1}{m} \sum_{i=1}^{m} \frac{1}{n} \sum_{j=1}^{n} \ell(c, h_i, k_{ij}, z_{ij}) \right| = \left| L_{\mathcal{D}}(c) - \frac{1}{m} \sum_{i=1}^{m} \ell(c, h_i, \mathbf{k}_i^n, \mathbf{z}_i^n) \right|.$$

Let $L^{c,n} := \ell_n(c, H, K^n, Z^n)$ with $H \sim \mathcal{D}_{\mathcal{H}}$ and $K^n, Z^n \sim (\mathcal{D}_{\mathcal{K}, \mathcal{Z}}^H)^n$. Since we know that $|L^{c,n} - \mathbb{E}L^{c,n}| \le 1$ and $\mathbb{E}[1/m \sum_i L_i^{c,n}] = L_{\mathcal{D}}(c)$, we can write, using Bernstein's inequality,

$$\mathbb{P}\left( |L_{\mathcal{D}}(c) - L_S(c)| > \epsilon \right) \le 2 \exp \left( -\frac{m\epsilon^2}{2\mathbb{V}(L^{c,n}) + \frac{2}{3}\epsilon} \right). \tag{8}$$

Based on (7), the law of total variance, and the law of large numbers, we can write

$$\mathbb{V}(L^{c,n}) = \mathbb{E}[\mathbb{V}(L^{c,n}|H)] + \mathbb{V}(\mathbb{E}[L^{c,n}|H]), \ H \sim \mathcal{D}_{\mathcal{H}}$$
$$= \mathbb{E}[\mathbb{V}(\ell(c, H, K, Z))|H)/n] + \mathbb{V}(\mathbb{E}[\ell(c, H, K, Z)|H]), \ H, K, Z \sim \mathcal{D}$$
$$= \frac{1}{n} \underbrace{\mathbb{E}[\mathbb{V}(\ell(c, H, K, Z))|H)]}_{\mathbb{E}[\mathbb{V}(\ell_c|H)]} + \underbrace{\mathbb{V}(\mathbb{E}[\ell(c, H, K, Z)|H])}_{\mathbb{V}(\mathbb{E}[\ell_c|H])}, \ H, K, Z \sim \mathcal{D}. \tag{9}$$

If we let $\mathbb{E}[\mathbb{V}(\ell_c|H)] := \mathbb{E}[\mathbb{V}(\ell(c, H, K, Z))|H)]$ and $\mathbb{V}(\mathbb{E}[\ell_c|H]) := \mathbb{V}(\mathbb{E}[\ell(c, H, K, Z)|H])$, we can rewrite (8) as

$$\mathbb{P}\left( |L_{\mathcal{D}}(c) - L_S(c)| > \epsilon \right) \le 2 \exp \left( -\underbrace{\frac{m\epsilon^2}{\frac{2}{n}\mathbb{E}[\mathbb{V}(\ell_c|H)] + 2\mathbb{V}(\mathbb{E}[\ell_c|H]) + \frac{2}{3}\epsilon}}_{\delta} \right). \tag{10}$$

To get the desired bound we only need to follow standard arguments in derivations of this type [9, for example], which we detail here for completeness. Equating the right-hand side of (10) to $\delta$, we have

$$\delta = 2 \exp \left( -\frac{m\epsilon^2}{\frac{2}{n}\mathbb{E}[\mathbb{V}(\ell_c|H)] + 2\mathbb{V}(\mathbb{E}[\ell_c|H]) + \frac{2}{3}\epsilon} \right). \tag{11}$$

Dividing both sides of (11) by 2 and taking the natural logarithm (as $\delta/2 > 0$), we get

$$\ln\left(\frac{\delta}{2}\right) = -\frac{m\epsilon^2}{\frac{2}{n}\mathbb{E}[\mathbb{V}(\ell_c|H)] + 2\mathbb{V}(\mathbb{E}[\ell_c|H]) + \frac{2}{3}\epsilon}$$

$$\implies \ln\left(\frac{2}{\delta}\right) = \frac{m\epsilon^2}{\frac{2}{n}\mathbb{E}[\mathbb{V}(\ell_c|H)] + 2\mathbb{V}(\mathbb{E}[\ell_c|H]) + \frac{2}{3}\epsilon}. \tag{12}$$

If we let $a = \ln\left(\frac{2}{\delta}\right)$ and $b = \frac{2}{n}\mathbb{E}[\mathbb{V}(\ell_c|H)] + 2\mathbb{V}(\mathbb{E}[\ell_c|H])$, and rearrange the terms, we can rewrite (12) as a quadratic equation in $\epsilon$,

$$m\epsilon^2 - \frac{2}{3}a\epsilon - ab = 0,$$

whose solutions are

$$\epsilon = \frac{a \pm \sqrt{a^2 + 9mab}}{3m}.$$

Since we are using $\epsilon$ to upper bound $|L_{\mathcal{D}}(c) - L_S(c)|$, we want the largest of the two solutions. We have $a > 0$ and $b \geq 0$, so $a^2 + 9mab > 0$. Thus, we take the solution with the positive sign. $\qquad\square$

**Proposition 2.** *For any $m, n > 0$ and any $\delta \in (0, 1]$, we have with probability at least $1 - \delta$ that*

$$L_{\mathcal{D}}(\tilde{c}^*) \leq L_{\mathcal{D}}(c^*) + 2\inf_\alpha\left[\frac{1}{3m}\left(g_\alpha + \sqrt{g_\alpha^2 + 18mg_\alpha\left(\frac{1}{n}\mathbb{E}[\mathbb{V}(\ell_{\mathcal{C}}|H)] + \mathbb{V}(\mathbb{E}[\ell_{\mathcal{C}}|H])\right)}\right) + 2\alpha\right],$$

*where $g_\alpha := \ln(2|\mathcal{C}_\alpha|/\delta)$.*

*Proof.* Define the semi-metric $d_\ell(c, c') := \max_{h\in\mathcal{H}, k\in\mathcal{K}, z\in\mathcal{Z}} |\ell(c(h, k), z) - \ell(c'(h, k), z)|$. Let $\mathcal{A}$ be the set of $\alpha$-nets over $\mathcal{C}$ according to $d_\ell$, that is,

$$\mathcal{A} := \{\mathcal{C}' \mid \mathcal{C}' \subseteq \mathcal{C}, \forall_{c\in\mathcal{C}}\exists_{c'\in\mathcal{C}'} d_\ell(c, c') \leq \alpha\}.$$

Let $\mathcal{C}_\alpha \in \arg\min_{\mathcal{C}'\in\mathcal{A}}|\mathcal{C}'|$. Clearly, this implies that

$$\forall_{c\in\mathcal{C}}\exists_{\hat{c}\in\mathcal{C}_\alpha}|L_{\mathcal{D}}(c) - L_{\mathcal{D}}(\hat{c})| \leq \alpha \text{ and } |L_S(c) - L_S(\hat{c})| \leq \alpha.$$

Note that $|\mathcal{C}_\alpha|$ is the covering number $N(\mathcal{C}, d_\ell, \alpha)$ [51]. Analogously to (8), we can write, for $\hat{c} \in \mathcal{C}_\alpha$:

$$\mathbb{P}\left(|L_{\mathcal{D}}(\hat{c}) - L_S(\hat{c})| > \epsilon\right) \leq 2\exp\left(-\frac{m\epsilon^2}{2\mathbb{V}(L^{\hat{c},n}) + \frac{2}{3}\epsilon}\right).$$

Define $\mathbb{V}(\ell_{\mathcal{C}}) := \sup_c \mathbb{V}(L^{c,n})$. Recalling that $\mathcal{C}_\alpha \subseteq \mathcal{C}$, it follows that

$$\forall_{\hat{c}\in\mathcal{C}_\alpha} \mathbb{P}\left(|L_{\mathcal{D}}(\hat{c}) - L_S(\hat{c})| > \epsilon\right) \leq 2\exp\left(-\frac{m\epsilon^2}{2\mathbb{V}(\ell_{\mathcal{C}}) + \frac{2}{3}\epsilon}\right).$$

If we apply the same argument in (9) to $\mathbb{V}(\ell_{\mathcal{C}})$, we can write

$$\forall_{\hat{c}\in\mathcal{C}_\alpha}\mathbb{P}\left(|L_{\mathcal{D}}(\hat{c}) - L_S(\hat{c})| > \epsilon\right) \leq 2\exp\left(-\frac{m\epsilon^2}{\frac{2}{n}\mathbb{E}[\mathbb{V}(\ell_{\mathcal{C}}|H)] + 2\mathbb{V}(\mathbb{E}[\ell_{\mathcal{C}}|H]) + \frac{2}{3}\epsilon}\right).$$

Applying the union bound over $\mathcal{C}_\alpha$, we get

$$\mathbb{P}\left(\exists_{\hat{c}\in\mathcal{C}_\alpha} : |L_{\mathcal{D}}(\hat{c}) - L_S(\hat{c})| > \epsilon\right) \leq \underbrace{2|\mathcal{C}_\alpha|\exp\left(-\frac{m\epsilon^2}{\frac{2}{n}\mathbb{E}[\mathbb{V}(\ell_{\mathcal{C}}|H)] + 2\mathbb{V}(\mathbb{E}[\ell_{\mathcal{C}}|H]) + \frac{2}{3}\epsilon}\right)}_{\delta}. \tag{13}$$

If we equate the right-hand side of 13 to $\delta$ and isolate $\epsilon$, as done in the proof of Proposition 1, we get

$$\epsilon_\alpha = \frac{1}{3m}\left[\ln\left(\frac{2|\mathcal{C}_\alpha|}{\delta}\right) + \sqrt{\ln\left(\frac{2|\mathcal{C}_\alpha|}{\delta}\right)^2 + 18m\ln\left(\frac{2|\mathcal{C}_\alpha|}{\delta}\right)\left(\frac{\mathbb{E}[\mathbb{V}(\ell_\mathcal{C}|H)]}{n} + \mathbb{V}(\mathbb{E}[\ell_\mathcal{C}|H])\right)}\right],$$
(14)

where we used the subscript in $\epsilon$ to note its dependency on $\alpha$. We have already shown that with probability at least $1-\delta$, we have that $\forall_{\hat{c}\in\mathcal{C}_\alpha} |L_\mathcal{D}(\hat{c}) - L_S(\hat{c})| \leq \epsilon_\alpha$. Given $c \in \mathcal{C}$, we can pick $\hat{c} \in \mathcal{C}_\alpha$ such that $d_\ell(c,\hat{c}) \leq \alpha$. Then,

$$\begin{aligned}
|L_\mathcal{D}(c) - L_S(c)| &= |L_\mathcal{D}(c) - L_S(\hat{c}) + L_S(\hat{c}) - L_S(c)| \\
&\leq |L_\mathcal{D}(c) - L_S(\hat{c})| + |L_S(\hat{c}) - L_S(c)| \\
&\leq |L_\mathcal{D}(c) - L_S(\hat{c})| + \alpha \\
&= |L_\mathcal{D}(c) - L_\mathcal{D}(\hat{c}) + L_\mathcal{D}(\hat{c}) - L_S(\hat{c})| + \alpha \\
&\leq |L_\mathcal{D}(c) - L_\mathcal{D}(\hat{c})| + |L_\mathcal{D}(\hat{c}) - L_S(\hat{c})| + \alpha \\
&\leq |L_\mathcal{D}(\hat{c}) - L_S(\hat{c})| + 2\alpha.
\end{aligned}$$
(15)

Thus, we can say that, with probability at least $1-\delta$, we have $\forall_{c\in\mathcal{C}} |L_\mathcal{D}(c) - L_S(c)| \leq \epsilon_\alpha + 2\alpha$. Since $\alpha$ was defined arbitrarily, we can write

$$\forall_{c\in\mathcal{C}} |L_\mathcal{D}(c) - L_S(c)| \leq \inf_\alpha(\epsilon_\alpha + 2\alpha).$$
(16)

Based on (16), we can write

$$L_\mathcal{D}(\tilde{c}^*) \leq L_S(\tilde{c}^*) + \inf_\alpha(\epsilon_\alpha + 2\alpha) \leq L_S(c^*) + \inf_\alpha(\epsilon_\alpha + 2\alpha) \leq L_\mathcal{D}(c^*) + 2\inf_\alpha(\epsilon_\alpha + 2\alpha).$$

$\square$

## A.2  Additional theoretical results

We now present additional theoretical results that complement those in the main paper. We start with a result upper bounding $|\mathcal{C}_\alpha|$, the measure of complexity used in Proposition 2.

**Proposition 3.** *Let $\alpha \in (0,1)$. Then, the smallest set $\mathcal{C}_\alpha \subseteq \mathcal{C}$ such that*

$$\forall_{c\in\mathcal{C}}\exists_{c'\in\mathcal{C}_\alpha}\forall_{h\in\mathcal{H},k\in\mathcal{K},z\in\mathcal{Z}} |\ell(c(h,k),z) - \ell(c'(h,k),z)| \leq \alpha$$

*has size at most $(1/\alpha)^{|\mathcal{H}||\mathcal{K}|}$.*

*Proof.* First, there exists a finite subset $\mathcal{M} \subseteq [0,1]$ of size $\alpha$ such that

$$\forall_{u\in[0,1]}\exists_{v\in\mathcal{M}}|u-v| \leq \alpha\,;$$

concretely, we may take $q = \lceil\frac{1}{2\alpha}\rceil < \frac{1}{\alpha}$, and let $\mathcal{M} = \{\frac{2i-1}{2q} : i \in \{1,\dots,q\}\}$.

Now, fix a classifier $c : \mathcal{H} \times \mathcal{K} \to \mathbb{R}$. We will specify a classifier $\hat{c} \in \mathcal{M}^{\mathcal{H}\times\mathcal{K}}$ with

$$|\ell(c(h,k),z) - \ell(\hat{c}(h,k),z)| < \alpha$$
(17)

for all $(h,k,z) \in \mathcal{H} \times \mathcal{K} \times \mathcal{Z}$. To do so, consider a tuple $(h,k) \in \mathcal{H} \times \mathcal{K}$. If $c(h,k) < 0.5$, we set $\hat{c}(h,k)$ such that $\ell(\hat{c}(h,k),0) \in \mathcal{M}$ is the closest value in $\mathcal{M}$ to $\ell(c(h,k),0)$, in particular within distance $\alpha$. Then, since the derivative of $\log$ is positive and decreasing, we also have that $|\ell(\hat{c}(h,k),1)-\ell(c(h,k),0)| \leq \alpha$. If instead $\ell(c(h,k),0) \geq 0.5$, we set $\hat{c}(h,k)$ so that $\ell(\hat{c}(h,k),1) \in \mathcal{M}$ is the closest value in $\mathcal{M}$ to $\ell(c(h,k),1)$, and the conclusion follows similarly. Thus, we have exhibited $\hat{c}$ with the property described in (17), and $\hat{c}$ is guaranteed to lie in a set of size at most $(1/\alpha)^{|\mathcal{H}||\mathcal{K}|}$, as required. $\square$

The analysis in Proposition 2 is based on the concept of covering numbers of function spaces, and Proposition 3 provides a simple upper bound for the covering number of $\mathcal{C}_\alpha$. It is also possible to derive results based on other measures of complexity of function spaces. We provide one illustrative example here, based on the notion of Rademacher complexity. This new result also assumes that the classifiers in $\mathcal{C}$ are logistic regression models.

**Proposition 4.** *Under the assumptions of Section 3, consider a hypothesis class $\mathcal{C}$ comprising logistic regression models over concatenated embeddings of user, prompt, and responses, with weights bounded in $L^2$ norm by $W$, and embedding $L^2$ norms bounded by 1. Let $c^*$ be the optimal model in this class for the population distribution $\mathcal{D}$, and let $\tilde{c}^*$ be the optimiser of the empirical loss. Then, with probability at least $1 - \delta$, we have*

$$L_{\mathcal{D}}(\tilde{c}^*) \leq L_{\mathcal{D}}(c^*) + \frac{2W}{\sqrt{m}} + 3\sqrt{\frac{g}{2m}} + \frac{1}{3m}\left[g + \sqrt{g^2 + 18gm\left(\frac{1}{n}\mathbb{E}[\mathbb{V}(\ell_{c^*}|H)] + \mathbb{V}(\mathbb{E}[\ell_{c^*}|H])\right)}\right],$$

*where $g = \log(6/\delta)$.*

*Proof.* First, note that the classification loss we are concerned with is a sum of non-i.i.d. terms:

$$L_S(c) = \frac{1}{m}\sum_{i=1}^{m}\frac{1}{n}\sum_{j=1}^{n}\ell(c(h_i, x_{ij}, y_{ij}, y'_{ij}), z_{ij}),$$

so we cannot immediately apply the classical methods of Rademacher complexity analysis (see, for example, Chapter 26 of [51]).

However, our data is i.i.d. at the level of the indices $i$, so we can begin by applying the framework of Rademacher complexity bounds at this level. We broadly follow the proof structure of [51, Theorem 26.5].

First, we have that

$$\sup_{c\in\mathcal{C}}(L_{\mathcal{D}}(c) - L_S(c)),$$

viewed as a function of $S$, satisfies the bounded-difference condition required for McDiarmid's inequality, from which it follows that this quantity concentrates around its expectation; with probability at least $1 - \delta$, we have

$$\sup_{c\in\mathcal{C}}(L_{\mathcal{D}}(c) - L_S(c)) \leq \mathbb{E}_S\left[\sup_{c\in\mathcal{C}}(L_{\mathcal{D}}(c) - L_S(c))\right] + \sqrt{\frac{\log(2/\delta)}{2m}}.$$

The key supporting result [51, Lemma 26.2] then allows us to relate the expected quantity on the right-hand side to the Rademacher complexity of our hypothesis class. Concretely, we have

$$\mathbb{E}_S\left[\sup_{c\in\mathcal{C}}(L_{\mathcal{D}}(c) - L_S(c))\right] \leq 2\mathbb{E}_{S'}[R(\ell\circ\mathcal{C}\circ S')],$$

where

$$\ell\circ\mathcal{C}\circ S' = \left\{\left(\frac{1}{n}\sum_{j=1}^{n}\ell(c(h_i, k_{ij}), z_{ij})\right)_{i=1}^{m} : c\in\mathcal{C}\right\}, \tag{18}$$

and $R$ denotes Rademacher complexity of the input set.

Now, the bounded-differences inequality required to employ McDiarmid's inequality applies to the Rademacher complexity as a function of $S'$ too, $R(\ell\circ\mathcal{C}\circ S')$, so that we may obtain with probability $1 - \delta$:

$$\mathbb{E}_{S'}[R(\ell\circ\mathcal{C}\circ S')] \leq R(\ell\circ\mathcal{C}\circ S) + \sqrt{\frac{\log(2/\delta)}{2m}}.$$

Putting all these parts together, yields

$$\sup_{c\in\mathcal{C}}(L_{\mathcal{D}}(c) - L_S(c)) \leq 2R(\ell\circ\mathcal{C}\circ S) + 3\sqrt{\frac{\log(4/\delta)}{2m}} \tag{19}$$

with probability at least $1 - \delta$.

Finally, denoting $c^*$ the optimiser of the true loss, and $\tilde{c}^*$ the optimiser of the empirical loss, we have

$$L_{\mathcal{D}}(\tilde{c}^*) - L_{\mathcal{D}}(c^*) = (L_{\mathcal{D}}(\tilde{c}^*) - L_S(\tilde{c}^*)) + (L_S(\tilde{c}^*) - L_S(c^*)) + (L_S(c^*) - L_{\mathcal{D}}(c^*))$$
$$\leq (L_{\mathcal{D}}(\tilde{c}^*) - L_S(\tilde{c}^*)) + (L_S(c^*) - L_{\mathcal{D}}(c^*)).$$

The first term is bounded by the inequality in Equation (19). The second term may be bounded by Bernstein's inequality: note that this deviates slightly from the approach set forward in [51, Chapter 26], but we can make use of Proposition 1 here. This ultimately results in a bound of the form:

$$L_{\mathcal{D}}(\tilde{c}^*) \leq L_{\mathcal{D}}(c^*) + 2R(\ell \circ \mathcal{C} \circ S) + 3\sqrt{\frac{g}{2m}} + \frac{1}{3m}\left[g + \sqrt{g^2 + 18gm\left(\frac{1}{n}\mathbb{E}[\mathbb{V}(\ell_{c^*}|H)] + \mathbb{V}(\mathbb{E}[\ell_{c^*}|H])\right)}\right]$$

where $g = \log(6/\delta)$.

Lastly, we derive an explicit form for the Rademacher complexity of our predictor, making use of the model class assumptions introduced in the statement of the result. To bound the empirical Rademacher complexity $R(\ell \circ \mathcal{C} \circ S)$, we can make a standard argument using several manipulations based on the Rademacher calculus, as described in [51, Chapter 26]. First, by the sum property, we focus on a single summand $j$ per dimension. Next, by the contraction lemma, and the fact that the logarithm-sigmoid composition is 1-Lipschitz, we may remove these elements from the set under consideration, revealing the linear function class that we hypothesise. With the assumptions on weight norms and embeddings made in the statement, we then obtain an overall Rademacher complexity of $W/\sqrt{m}$. This results in an overall bound of:

$$L_{\mathcal{D}}(\tilde{c}^*) \leq L_{\mathcal{D}}(c^*) + \frac{2W}{\sqrt{m}} + 3\sqrt{\frac{g}{2m}} + \frac{1}{3m}\left[g + \sqrt{g^2 + 18gm\left(\frac{1}{n}\mathbb{E}[\mathbb{V}(\ell_{c^*}|H)] + \mathbb{V}(\mathbb{E}[\ell_{c^*}|H])\right)}\right],$$

as required. □

## B  Details of the experiments and additional empirical results

We tried to keep our experimental setup as simple as possible to provide a realistic estimate of out-of-the-box performance. In particular, we did not carry out an extensive search over network architectures or hyper-parameters, keeping most of them at sensible defaults from the outset. We also did not specialise the training procedure to RFM's architecture, using standard gradient descent to minimise the training and adaptation losses.

As mentioned in the main text, we used Google DeepMind's [26] Gemma 1.1 2B to implement RFM and the baselines. The maximum context and response lengths were set to $l_x = l_y = 1,525$ tokens. Training and adaptation were carried out using gradient descent with a learning rate of $10^{-5}$. We performed a random 90%–10% split of the training set and used the error in the smaller subset (a validation set) as a criterion to select the model to undergo adaptation.

### B.1  Empirical analysis

**Problem setup.**  The following 13 features were used:

- $\phi_1(x, y)$: the length of $y$.
- $\phi_2(x, y)$: the average sentence length in $y$.
- $\phi_3(x, y)$: the average word length in $y$.
- $\phi_4(x, y)$: the proportion of characters in $y$ that are vowels.
- $\phi_5(x, y)$: the proportion of characters in $y$ that are punctuation symbols.
- $\phi_6(x, y)$: the proportion of transitions between words in $y$ that are alliterations.
- $\phi_7(x, y)$: the proportion of words in $y$ that are adjectives.
- $\phi_8(x, y)$: the proportion of words in $y$ that are adverbs.
- $\phi_9(x, y)$: the proportion of words in $y$ that are verbs.
- $\phi_{10}(x, y)$: the proportion of words in $y$ that are nouns.
- $\phi_{11}(x, y)$: the proportion of words in $y$ that are synonyms of one of the words in $x$.
- $\phi_{12}(x, y)$: the proportion of words in $y$ that are antonyms of one of the words in $x$.
- $\phi_{13}(x, y)$: the proportion of words in $y$ that also appear in $x$.

All the features were normalised to fall in the interval $[0, 1]$ and then centered around their median value so that positive values represent values above the median (and vice versa). Our features were designed to capture potentially conflicting subjective criteria, yet they remain inherently neutral (*i.e.*, possessing no inherent valence). These features also vary in how they access inputs $x$ and $y$, which consequently influences their anticipated learning difficulty. We identify three categories of features. "Structural" features (*e.g.*, $\phi_1, \phi_2, \phi_3$) treat $x$ and $y$ as raw strings devoid of meaning and are thus expected to be straightforward to learn. In contrast, "syntactic" features (*e.g.*, $\phi_7, \phi_8, \phi_9$) analyze the grammatical role of words in a sentence, and their dependence on inter-word relationships makes them more challenging to learn. Finally, "semantic" features (*e.g.*, $\phi_{11}, \phi_{12}$), which rely on the meaning of words, are likely the most difficult to learn.

As discussed in Section 5, we defined a user as a vector $\boldsymbol{\omega} \in \{-1, 1\}^{13}$. Given an example $(x, y, y')$ and a user $\boldsymbol{\omega}$, the corresponding preference $z$ was determined as

$$z = \mathbb{1}\{\langle \boldsymbol{\phi}(x, y), \boldsymbol{\omega} \rangle > \langle \boldsymbol{\phi}(x, y'), \boldsymbol{\omega} \rangle\}, \tag{20}$$

where $\mathbb{1}\{\cdot\}$ is the indicator function and $\{\phi_i(x, y)\}_{i=1}^{13}$ are the features described above.

We defined a distribution over $\mathcal{H}$, $\mathcal{D}_{\mathcal{H}}$, parameterised by a single parameter $p$ determining $\mathbb{P}(\omega_i = 1)$ for each $i = 1, 2, ..., 13$ independently. That is, in order to sample a user $h$ from $\mathcal{D}_{\mathcal{H}}$, for each $i = 1, 2, ..., 13$ we sample $Z \sim \text{Bernoulli}(p)$ and set $\omega_{hi} = 2Z - 1$. The parameter $p$ of the distribution $\mathcal{D}_{\mathcal{H}}$ allows us to control how homogeneous in terms of preferences groups sampled from it tend to be. The entropy of $\mathcal{D}_{\mathcal{H}}$ peaks at $p = 0.5$, resulting in the maximum degree of disagreement. As $p \to 0$ or $p \to 1$, the preferences become more homogeneous. Figure 5 illustrates the effect of $p$ on the homogeneity of the users' preferences.

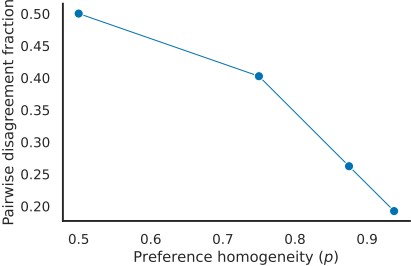

Figure 5: Pairwise user disagreement as a function of $p$. For each value of $p \in \{0.5, 0.75, 0.875, 0.9375\}$, we sampled 100 users from $\mathcal{D}_{\mathcal{H}}$ and computed the fraction of the examples in the UltraFeedback training set on which they disagreed.

We used sets of raters $\hat{\mathcal{H}}$ of different sizes: $m \in \{20, 40, 60, 80, 100\}$. Each run used a different set $\hat{\mathcal{H}}$ with raters sampled from $\mathcal{D}_{\mathcal{H}}$ as explained above. We also sampled a set of 500 held-out users per experiment using the same procedure. The held-out users were used to assess the inter-user generalisation of the models.

Unless otherwise noted, the default values for the parameters used in the experiments were: $m = 60$ raters, preference homogeneity level $p = 0.7$, and $\hat{n} = 30$ examples used for adaptation.

**Training and adaptation.** Training was carried out for $6,000$ parameter updates with a batch size of 32. This means that the training procedure went over the entire UltraFeedback training set approximately three times. Each time the example $(x_i, y_i, y'_i,)$ was encountered, a new rater $\hat{h}$ was sampled uniformly at random from $\hat{\mathcal{H}}$ and the preference $z_i$ was determined through (20) with the corresponding $\boldsymbol{\omega}_{\hat{h}}$. For the experiments shown in Figure 1a specifically—with a varying number $m$ of raters—we needed to make sure that each rater was trained with roughly the same number of examples $n$. So, we extended training proportionally to $m$ (Figure 6 shows the validation and test errors along training in this experiment).

To perform the adaptation, we sampled $\hat{n} \in \{10, 30, 50, 70, 90\}$ examples from UltraFeedback's training set uniformly at random. Analogously to training, each time the example $(x_i, y_i, y'_i,)$ was encountered a user $h$ was sampled uniformly at random from the set of 500 held-out users, with

the corresponding preference $z$ determined through (20). We carried out a total of $6,000$ parameter updates with a batch of 32 for all 500 held-out users combined (so, only around 384 updates per user, in expectation).

For each experiment, we carried out training followed by adaptation 5 times. All the numbers reported are averages over the corresponding 5 runs. The metric we report for adaptation in Figures 1 and 3 is the *inter-user test accuracy*, an estimate of the fraction of examples in the test set correctly classified by the models, per user. To compute it, we went over the test set 50 times, always assigning to each example a user sampled uniformly at random from the set of 500 held-out users. This means that each user is evaluated in approximately 100 examples, in expectation. The accuracy reported in Figures 1 and 3 is the average number of correctly classified examples over the 50 passes over the test set.

**Additional results.**    Figures 6 and 7 show the baseline and RFM's accuracy on the test and validation sets during training. In Figure 6 we see the effect of varying the number of raters $m$, while in Figure 7 we see the effect of varying the preference homogeneity parameter $p$. The values reported are the *intra-user validation accuracy* and the *intra-user test accuracy*. Their computation is analogous to that of the *inter-user test accuracy* explained above, with held-out users replaced by raters and the test set replaced by the validation set when applicable. Note that, in contrast with the adaptation error shown in Figure 1, the training error tends to go up with the number of raters $m$. This makes sense, since each rater corresponds to a classification problem being solved in parallel.

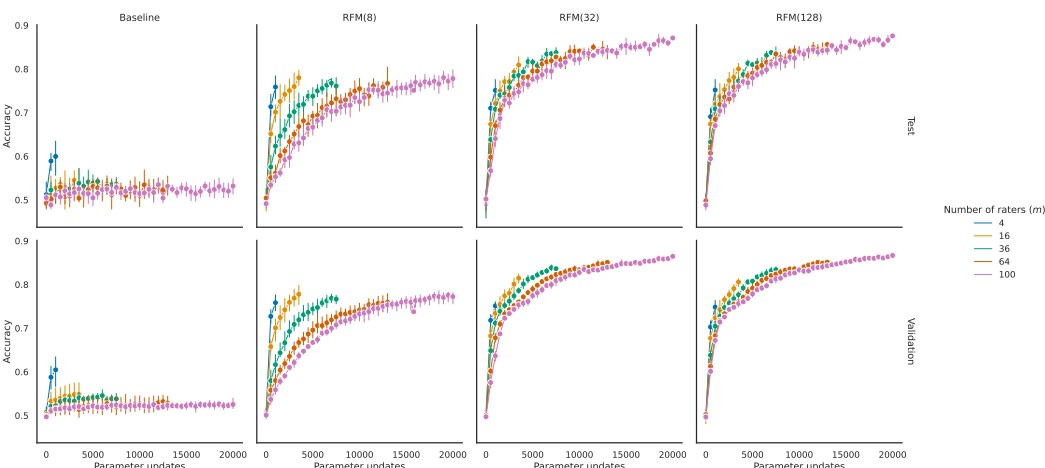

Figure 6: Validation and test accuracies as a function of update steps and number $m$ of training raters during the training phase (estimate of intra-user generalisation). The number of parameter updates is proportional to $m$ to ensure that all raters see roughly the same number $n$ of training examples. Error bars are $99\%$ confidence intervals over 5 runs.

Figure 1c shows the performance of the baseline and RFM when using $\hat{n} \in \{10, 30, 50, 70, 90\}$ examples for adaptation. Although these are small numbers from a learning perspective, one may ask what happens under even more stringent conditions. To answer this question, we extended the RFM(32) results in Figure 1c to include test accuracies with $\hat{n} \in \{1, 3, 5\}$. The results are shown in Table 1, together with RFM(32)'s results from Figure 1c.

As expected, RFM's performance improves monotonically with the number of examples $\hat{n}$ used for adaptation. Note that even the results with a single example remain above random chance (though significantly lower than the results with $\hat{n} \geq 10$).

We point out that worse performance with very few adaptation examples is expected, for it reflects the intrinsic difficulty of the learning problem. Given that each user's preferences are derived from a combination of 13 features, many different combinations of those 13 features may explain the preference behind a few training examples, and multiple training examples may be necessary to disambiguate. It is reassuring to see that RFM is still able to capture some of the problem structure under these extreme conditions, and that its performance monotonically increases with $\hat{n}$.

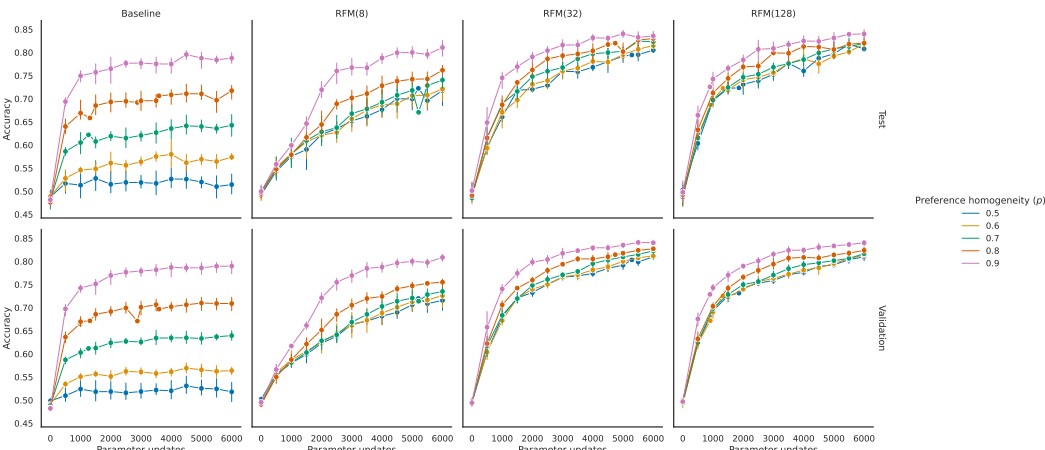

Figure 7: Validation and test accuracies as a function of update steps and preference heterogeneity $p$ during the training phase (estimate of intra-user generalisation). Error bars are $99\%$ confidence intervals over 5 runs.

Table 1: Accuracy in predicting the preferences of $500$ held-out users on UltraFeedback's test set after adaptation (cf. Figure 1). The range of values shown are $99\%$ confidence intervals over 5 runs.

| $\hat{n}$ | Test accuracy |
|---|---|
| 1 | $0.5481 \pm 0.0217$ |
| 3 | $0.5749 \pm 0.0109$ |
| 5 | $0.5891 \pm 0.0056$ |
| 10 | $0.7053 \pm 0.0182$ |
| 30 | $0.7657 \pm 0.0790$ |
| 50 | $0.7770 \pm 0.0333$ |
| 70 | $0.7835 \pm 0.0217$ |
| 90 | $0.7900 \pm 0.0172$ |

In Figure 1b we show the effect of varying the level of homogeneity $p$ of the distribution $\mathcal{D}_{\mathcal{H}}$ from which raters and held-out users are sampled. In some real scenarios, there may be a discrepancy between the distribution used to sample raters and the real distribution underlying users. To simulate this scenario, we ran an experiment in which held-out users were sampled from a fixed distribution with $p = 0.65$ while raters were sampled from distributions with varying $p$. Results are shown in Figure 8. The discrepancy between the rater and held-out user distributions has a negative impact on both RFM's and the baseline's performance, as expected. However, the negative impact on the baseline is much more severe, and, in contrast with RFM, it increases as $p \to 0.5$.

## B.2 Comparisons

**Comparison with the linear baseline.** The linear baseline was adapted on top of the trained baseline using gradient descent with a learning rate of $10^{-5}$ (the same used for RFM). We froze all the parameters of the baseline trained with $m = 60$ raters and preference homogeneity $p = 0.7$ (*c.f.* Figure 1), except for the last layer. Then, we replaced the last layer with $500$ linear layers, one for each held-out users $h$, and trained them using the data in the corresponding $S_h$. Note that this comes down to performing $500$ logistic regressions in parallel on top of the frozen baseline's user-agnostic features (whose dimension is $1024$). Adaptation followed the exact same protocol adopted for RFM, with $6,000$ parameter updates using a batch size of $32$ shared among all users.

As an aside, note that using the protocol above to train all 2 billion parameters of the baseline $500$ times would be infeasible. This illustrates our point in Section 3 advocating a small number $d$ of adaptable parameters.

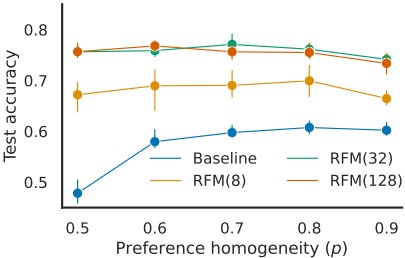

Figure 8: Accuracy in predicting the preferences of 500 held-out users on the test set after adaptation (estimate of inter-user generalisation). Training was carried out with $m = 60$ raters and $\hat{n} = 30$ examples were used for adaptation. The homogeneity parameter was fixed at $p = 0.65$ for the 500 held-out users, and varied for the training raters as shown on the $x$-axis. Error bars are 99% confidence intervals over 5 runs.

.

**Comparison with the non-linear baseline.** Park et al. [42] and Zhong et al. [64] propose principled (and similar) methods to adapt a reward model to individual users. Since the former is slightly more general, we compared RFM against it.

Using our terminology, we can describe Park et al.'s [42] method as having a single learning phase that is intermediate between training and adaptation. If we partition the shared parameters $\boldsymbol{\theta} = [\boldsymbol{\theta}_1, \boldsymbol{\theta}_2]$, the method consists in keeping $\boldsymbol{\theta}_1$ fixed while $\boldsymbol{\theta}_2$ is learned together with the vectors $\mathbf{w}_h$ associated with users. We note that, unlike RFM, this approach requires that the vectors $\mathbf{w}_h$ are simultaneously learned for all users $h \in \mathcal{H}$ we are interested in adapting the model to.

Following Park et al.'s [42] experiments, we froze the backbone Gemma model ($\boldsymbol{\theta}_1$), represented $\phi$ as a multilayer perceptron (MLP) with 3 or 5 hidden layers containing 32 units each ($\boldsymbol{\theta}_2$), and learned the MLP together with the vectors $\mathbf{w}_h$. To make sure $\boldsymbol{\theta}_1$ was initialised with reasonable values, we ran our training phase with $\boldsymbol{\theta}_1, \boldsymbol{\theta}_2$, and $\mathbf{W}$ being learned together, where the rows of $\mathbf{W}$ correspond to the raters $\hat{h} \in \hat{\mathcal{H}}$ (cf. Equation (6)).

As shown in Figure 3, the non-linear baselines did not perform well. We conjectured that this may be due an excessive number of parameters to be learned with $\hat{n} = 10$ examples, so we re-ran the experiment using $\hat{n} = 100$. Results are shown in Table 2 together with the other results from Figure 3. Although the non-linear baselines perform slightly better with larger $\hat{n}$, they are still outperformed by the non-adaptive and linear baselines, and considerably outperformed by RFM.

Table 2: Detailed comparison of RFM against linear and non-linear baselines (cf. Figure 3). The values shown are the accuracy in predicting the preferences of 500 held-out users using $\hat{n}$ examples for adaptation, together with 99% confidence intervals over 5 runs.

| Model | $\hat{n}$ | Test accuracy |
|---|---|---|
| Baseline | 10 | $0.6258 \pm 0.0246$ |
| Linear baseline | 10 | $0.6285 \pm 0.0176$ |
| Non-linear $\phi$ (3 layers) | 10 | $0.5543 \pm 0.0381$ |
| | 100 | $0.5706 \pm 0.0207$ |
| Non-linear $\phi$ (5 layers) | 10 | $0.5237 \pm 0.0273$ |
| | 100 | $0.5620 \pm 0.0452$ |
| RFM(32) | 10 | $0.7053 \pm 0.0182$ |

An alternative to the non-linear architecture proposed by Park et al. [42] would be to have a separate MLP per user (that is, we replace $\mathbf{w}_h$ with $\boldsymbol{\theta}_h$). This architecture would probably required even more examples to be adapted, since the MLPs' parameters $\boldsymbol{\theta}_h$ would be trained with the data in $S_h$ only.

**Comparison with in-context methods.** We present the prompt used to evaluate the in-context capabilities of Gemini 1.5 Pro and GPT-4o in Figure 9. It has three main parts: (i) the system instructions, delimited by [System] and [End of system]; (ii) a sequence of 10 previous user

ratings (whose formatting is described in the system instructions); and (iii) the prompt, first response, and second response to be assessed by the LLM. To avoid positional biases, we have the LLM assess the first and second responses both in their original order and in reversed order and average over the correctness of both LLM outputs.

For each example in UltraFeedback's test set, we sample one user $h$ uniformly at random from the set of 500 held-out users. We then draw from the training set 10 examples of responses previously ranked by $h$. The previously ranked responses are inserted in part (ii) of the prompt template, and the test example itself is inserted in part (iii) of the prompt template. We then compare the predicted comparison outcome against user $h$'s ranking for that test example.

We also evaluate Gemini's zero-shot agreement with the held-out users (*Gemini (zero-shot)*) to help assess whether adding previously ranked responses as context makes a difference in terms of performance. In that setting, we omit the paragraph starting with "We will provide a few examples" in part (i) of the prompt template and remove part (ii) altogether.

The fact that the LLMs' performance essentially reduces to chance in Figure 3 is somewhat surprising. We hypothesise the explanation is twofold: 1) despite the information being available in the prompt, the LLMs do not take advantage of the previous ratings to inform their decisions and instead revert to making a "judgement call" based on their existing alignment; and 2) this alignment is somewhat orthogonal to the features $\{\phi_i\}_{i=1}^{13}$ we used (described in Appendix B.1). Preliminary experiments with the features provided with the UltraFeedback dataset corroborate this hypothesis. The UltraFeedback dataset comes with four features computed using OpenAI's [39] GPT-4: *helpfulness*, *honesty*, *instruction-following*, and *truthfulness*. Each example in the dataset has a score between 1 and 5 associated with each feature. We defined a user whose preferences were fully determined by the *helpfulness* features and reran the experiment described above. Results are in Table 3 and demonstrate that 1) just like with our main in-context comparison experiment shown in Figure 3, the effect of 10-shot in-context prompting is negligible in comparison to zero-shot prompting, and 2) the LLMs perform considerably above chance when trying to predict preferences induced by a feature computed itself by an LLM.

Table 3: Accuracy in predicting the preferences of a held-out user induced by the feature *helpfulness* provided with the UltraFeedback dataset. The three models used $\hat{n} = 10$ examples for adaptation.

| Model | Test accuracy |
|---|---|
| Gemini 1.5 Pro (zero-shot) | 0.6341 |
| Gemini 1.5 Pro (in-context, 10-shot) | 0.6199 |
| GPT-4o (in-context, 10-shot) | 0.6392 |

We considered using the UltraFeedback features in our experiments, but for a more rigorous study we needed features that could be computed unequivocally and efficiently, and thus easily extrapolated beyond the UltraFeedback dataset. This was essential for the experiments with best-of-$n$ shown in Figure 2, for example, in which we had to compute the features associated with examples $(x, y, y')$ not in the UltraFeedback dataset.

**Comparison with VPL.** Poddar et al.'s [44] evaluation protocol for VPL allows to measure intra-user generalization, but not inter-user generalization. This is because the preferences used for evaluation are obtained from the same four raters that were used to train the model. Those raters' preferences are derived from the four UltraFeedback features mentioned above: *helpfulness*, *honesty*, *instruction-following*, and *truthfulness*, with each rater focusing exclusively on a single feature. In terms of evaluating intra-user generalization, VPL relies on an episodic protocol: for every rater $\hat{h}_i$ and for every tuple $(x, y, y', z)$ in the rater's test set, the authors simulate a new adaptation problem by drawing between two to eight other tuples sampled at random from the rater's test set to provide as context for the inference network to make a prediction. This means that the rater's identity $\hat{h}_i$ is never explicitly revealed to the model, but only contextually through the two to eight other tuples. In contrast, our evaluation methodology is more akin to transfer learning evaluation: for each rater, we adapt RFM on a held-out set of examples. The *same* held-out set is used for all test examples, that is, we learn $\mathbf{w}$ once using the held-out data and then use it to process all the test tuples $(x, y, y', z)$.

```
[System]
Please act as an impartial judge and evaluate the quality of the responses provided
by two AI assistants to the user question displayed below.

A rating starts with the tag [User question] followed by the context and the tag
[End of user question]. After that, we have the tag [The Start of Model A's answer]
followed by the first response. The first response ends with the tag [The End of
Model A's answer]. We then have the tag [The Start of Model B's answer] followed by
the second response. The second response ends with the tag [The End of Model B's
answer].

We will provide a few examples of previous ratings to help you understand the task.
The example ratings will have the structure above followed by the tag [Verdict],
the verdict ("[[A]]" if assistant A is better, and "[[B]]" if assistant B is
better), and the tag [End of verdict].

We are interested in your evaluation of the last two responses. Begin your
evaluation by comparing these two responses and provide a short explanation. Do not
favor certain names of the assistants. Be as objective as possible. After providing
your explanation, output your final verdict by strictly following this format:
"[[A]]" if assistant A is better, and "[[B]]" if assistant B is better.
[End of system]

[User question]
<...>
[End of user question]

[The Start of Model A's answer]
<...>
[The End of Model A's answer]

[The Start of Model B's answer]
<...>
[The End of Model B's answer]

[Verdict]
<...>
[End of verdict]

<...>

[User question]
<...>
[End of user question]

[The Start of Model A's answer]
<...>
[The End of Model A's answer]

[The Start of Model B's answer]
<...>
[The End of Model B's answer]
```

Figure 9: The prompt template used for in-context evaluation. When evaluating Gemini's zero-shot capabilities we omit the paragraph starting with "We will provide a few examples" in the prompt.

The VPL authors provide the exact data used for evaluation on UltraFeedback.[3] For simplicity, we adapt our evaluation protocol so as to get an unbiased estimate of the episodic metric used to evaluate VPL. We hold out 250 examples from VPL's training set to sample adaptation sets. After RFM has been trained on the four UltraFeedback raters derived from UltraFeedback features, we loop over raters, number $\hat{n} \in \{2, 3, 4, 5, 6, 7, 8\}$ of examples used for adaptation, and 5 random seeds, each time drawing $\hat{n}$ examples from the held-out set for that rater, discarding the pre-trained set of weights $\mathbf{w}$ for that rater, and learning a new $\mathbf{w}$ on the sampled adaptation set. We then aggregate the accuracies measured on the test set across adaptation set sizes, raters, and random seeds (see Figure 10 for non-aggregated results). As a result, the examples used for adaptation for each individual test example do not correspond exactly to the ones used by VPL (and are shared across test examples), but they are equally disjoint from the training set and their distribution is i.i.d. with respect to the distribution VPL samples from.

We obtain an averaged test accuracy of 61.61% against VPL's 61.49%. Given the methodological caveats outlined above, the conclusion we draw is that, despite its simplicity, RFM achieves an intra-user generalization performance comparable with VPL's. We again highlight that this notion of generalisation is distinct from inter-user generalisation, which is the main one we are targeting in this work, but for which VPL does not provide metrics to compare against.

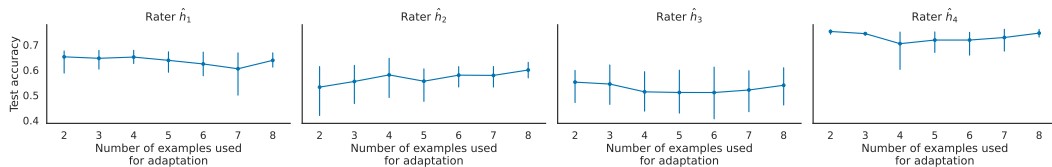

Figure 10: Test accuracies for RFM(32) on the UltraFeedback-derived UF-P-4 task introduced by Poddar et al. [44], broken down by rater and number of examples used for adaptation. The error bars represent 99% confidence intervals over 5 randomly-sampled adaptation sets.

## B.3 Modelling groups of real users

In the experiments described in Appendices B.1 and B.2 we had control over the definition of the training raters and held-out users. In the experiments with reward models acting as users we removed this assumption. As mentioned in Section 5, we performed our experiments with two datasets: UltraFeedback and PersonalLLM. We now describe each in turn.

For the experiments with UltraFeedback, shown in Figure 4a, we used the following 8 publicly-available reward models to emulate raters and users:

- OpenAssistant_reward-model-deberta-v3-large-v2,
- weqweasdas_RM-Mistral-7B [20, 59],
- OpenAssistant_oasst-rm-2.1-pythia-1.4b-epoch-2.5,
- Ray2333_GRM-Gemma-2B-sftreg [60],
- Ray2333_reward-model-Mistral-7B-instruct-Unified-Feedback [60],
- weqweasdas_RM-Gemma-7B [20],
- internlm_internlm2-7b-reward [23],
- openbmb_Eurus-RM-7b [62].

All the models above are available on the Hugging Face website.[4] For each reward model $r_k$ and each example $(x_i, y_i, y_i')$ in the training and test sets, we defined $z_i^k = \mathbb{1}\{r_k(x_i, y_i) > r_k(x_i, y_i')\}$.[5] We performed leave-one-old cross validation using the 8 resulting raters, as explained in Section 5. As before, training was carried out for $6,000$ parameter updates with a batch size of 32. We performed

---

[3]github.com/WEIRDLabUW/vpl_llm?tab=readme-ov-file#data-and-pretrained-models

[4]huggingface.co.

[5]The resulting data is available at huggingface.co/datasets/google/rfm-rm-as-user-dataset.

2 training runs followed by 5 adaptation runs each, totalling 10 runs. Figure 11 shows the detailed results obtained in our experiments with the UltraFeedback dataset (Figure 4 is a slice of this figure with the number of adaptation examples fixed at $\hat{n} = 50$).

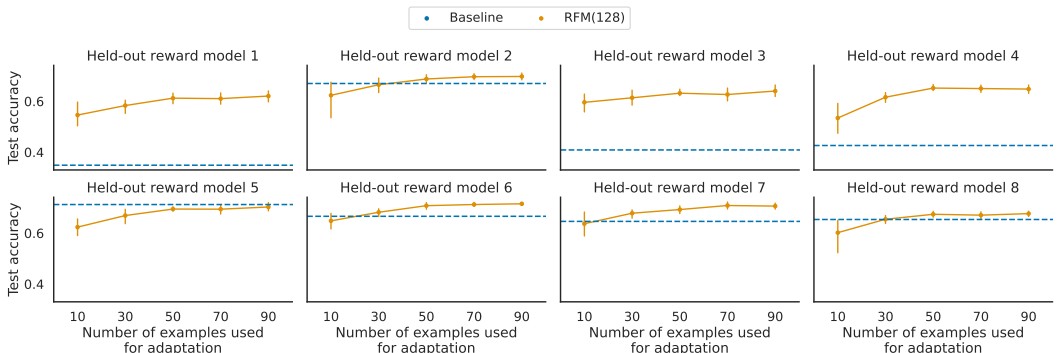

Figure 11: Accuracy in predicting the preferences of held-out "reward-models users" on test set after adaptation (estimate of inter-user generalisation). Error bars are $99\%$ confidence intervals over 10 runs (5 adaptation runs on top of 2 training runs).

We now describe our experiments with the PersonalLLM dataset, whose results are shown in Figure 4b. Each context $x$ in this dataset has 8 responses, and each response has been scored by 10 reward models. For our experiments, we picked the first two responses to each context $x$ and used them as our pair $(y, y')$. We then proceeded exactly as in the experiments with the UltraFeedback dataset, except that we performed 5 runs each involving training followed by adaptation.

## C  Expanded discussion of related work

In this section we present a more detailed discussion on the works we consider to be more directly relevant to ours.

Shenfeld et al. [52] concurrently propose the same architecture as RFM, but under a different name: "PReF", for *personalisation via reward factorization*. They also approach the subject from a different angle: while we study the learning problem from a higher level of abstraction, and then specialise our results to RFM, they focus on how to exploit RFM's simple architecture to improve several aspects of the training pipeline. Among these, two contributions stand out: a stable initialisation of the model through singular value decomposition and an efficient way of selecting examples for adaptation via active learning. Their theoretical results regard the latter. Shenfeld et al. [52] also present a thorough empirical evaluation of their method, including an ablation study of their proposed improvements, comparisons with Chen et al.'s [13] and Poddar et al.'s [44] approaches (discussed below), and a small (but interesting) experiment involving human users.

Bose et al. [8] also concurrently propose an architecture very similar to RFM, this time under the name *LoRe* (for "low-rank reward modelling"). The LoRe architecture closely resembles RFM's, in that both express individual reward functions as linear combinations of learned reward features (or "reward basis" in LoRe's terminology). A small, but potentially relevant difference is that their vector of coefficients $\mathbf{w}$ is a distribution (that is, $\sum_{i=1}^{d} w_i = 1$ and $0 \leq w_i \leq 1$ for all $i = 1, 2, ..., d$). This slightly restricts the expressiveness of the model. Bose et al. [8] do not present theoretical results, but their empirical evaluation of the proposed architecture is extensive. They make some of the same comparisons as us—namely: baseline, linear baseline, and VPL—and also compare the proposed architecture with that of Chen et al. [13], discussed below.

Zhong et al. [64] and Park et al. [42] also propose methods that are reminiscent of RFM. Both papers put more emphasis on the theoretical analysis than on empirical results. Zhong et al. do not present experiments, while Park et al. only present simple empirical evaluations. We briefly describe one instantiation of Park et al.'s approach in Section 5 and Appendix B.2, and also compare it with RFM.

Go et al.'s [25] *compositional preference model* (CPM) expresses the learned reward function as a linear combination of features computed from the context and response. However, unlike RFM—which learns features using (1)—, the CPM approach uses handcrafted ordinal features computed by LLMs using pre-specified prompts.

Chen et al. [13] propose to replace the Bradley-Terry model with Coombs's [17] *ideal point model*. This model posits that people make assessments of a given object based on the distance between the representation of the object in a latent space and an "ideal reference point" in the same space. While the inner product between RFM's $\phi_\theta(x, y)$ and $\mathbf{w}_h$ does not strictly qualify as a distance function, we can think of $\mathbf{w}_h$ as being akin to an "ideal point" for user $h$ against which $\phi_\theta(x, y)$ is compared. Chen et al. [13] propose two different instantiations of the ideal point model—Model A and Model B—which differ from RFM in a few aspects. Like RFM's $\phi_\theta(x, y)$, Model A computes features from the context and response jointly, but the ideal point is constructed as a convex combination of "prototypical" ideal points and the comparison is made using the Euclidean distance. Model B defines an ideal point that depends on the context only, also as a convex combination of prototypical ideal points. Unlike RFM, Model B computes response features separately and compares them to the context ideal point using the cosine similarity. Both Model A and Model B constrain the user-specific parameters to lie on a simplex whose number of vertices becomes an extra hyper-parameter (in addition to the dimension $d$ of the ideal points). This presents an additional optimisation challenge and, as the authors themselves point out, does not generalise to users whose ideal point would fall outside of the convex hull.

Poddar et al.'s [44] *variational preference learning* (VPL) works by encoding tuples $\{x_i, y_i, y_i', z_i\}_{i=1}^m$ associated with a user $h$ into a latent variable, and then conditioning the reward model (and policy) on the latent variable. Zhao et al. [63] cast the problem of personalised preference learning as a few-shot learning problem and tackle it with in-context meta-learning. Their *group preference optimization* (GPO) approach trains a transformer to predict target preferences for a given homogeneous group of users from a context set of preferences from the same group. Like RFM, both works learn from a context set of preferences from a new user (or group of users). Unlike RFM—which learns through optimizing $\mathbf{w}_h$—both works amortize the process using a neural network (namely, VPL's encoder and GPO's transformer).

Li et al. [36] propose a personalised RLHF (P-RLHF) framework in which a learnable user model computes an embedding for each user as the concatenation of an implicit embedding (that depends on the user's unique identifier) and an optional explicit embedding (that depends on textual information about the user). The user embedding is then used to condition a base LLM through soft prompting. New users are accommodated by using a generic implicit embedding and computing an explicit embedding based on the new user's textual information (if available).

Dumoulin et al. [22] present results on "rater misspecification", in which a model trained on the preferences of a single rater which favors two characteristics (e.g., short generations or long generations, but not middling generations) behaves very differently from a model trained on the preferences of two raters, each of which favor one characteristic (short generations for one rater, long generations for the other). The authors frame the issue from the perspective of a misspecified generative process for pairwise preferences and show that under correct assumptions (namely, the preference dataset is a mixture of individual raters' preferences) it is possible to accurately capture multiple raters' preferences. In practice, the solution the authors propose (explicitly introducing as many models as there are raters) works in synthetic settings but was never shown to work at scale on larger problems. It assumes that one knows the number of raters in the preference dataset, but it does not assume that each individual preference tuple is annotated with the rater ID. The solution also does not consider how one would accommodate new users.

Chakraborty et al. [12] propose to learn a user-specific reward function by using an expectation-maximisation algorithm to define a mixture of preference distributions. To learn a single policy that better represent diverse human preferences, the authors propose a *MaxMin* alignment objective inspired by the Egalitarian principle in social choice theory.

# D  Broader impact

Beyond an improved user experience, adaptive reward models have great potential for positive impact. For example, they may allow for the inclusion of minority opinions and the representation of diverse

viewpoints in model outputs. However, as with most technological advances, this does not come without risks. If not implemented with ethical implications in mind, the specialisation of LLMs to user preferences may result in models behaving in undesirable ways. It may also reinforce existing points of view through sycophantic behaviour, contributing to the polarisation of opinions and the creation of "echo chambers".

There are ways to anticipate and mitigate these undesirable outcomes on the methodological, technical, and societal fronts. Methodologically, it is important to define rubrics for data collection and curation that clearly distinguish between genuine disagreement on subjective matters and denial of facts or deviations from prevalent ethical norms and scientific consensus. From a technical standpoint, the fact that a user-specialised reward model will be used to modulate the outputs of an LLM renders the LLM itself a safeguard mechanism. For example, the best-of-$n$ approach used in this paper re-ranks the LLM's outputs based on the adapted reward model. Consequently, if the LLM generates factually correct and ethical text, the re-ranking primarily prioritises or de-prioritises aspects of a subjective nature—like the style of the prose, for instance. Finally, on a societal level, the personalisation of LLMs should be part of, and would directly benefit from, the wider ongoing discussion regarding the deployment of this new technology.

On the positive side, adaptive reward models and the consequent personalisation of LLMs offer a great opportunity for the inclusion of diverse points of view into model outputs. Current reward models reflect the average preferences of the target population, which excludes under-represented or "outlier" preferences. In using specialised reward models to adapt LLM outputs to individuals, we create systems that can more accurately and reliably reflect the perspectives of users who hold minority views, potentially empowering them, together with everyone else, to participate more fully in social debate.

