# OpenReview forum: "Capturing Individual Human Preferences with Reward Features"
_NeurIPS.cc/2025/Conference — NeurIPS 2025 poster_

### Official Review · Reviewer_DjQY · 2025-06-29

**Clarity:** 3
**Significance:** 3
**Originality:** 3
**Rating:** 4
**Confidence:** 4

**Summary:**

The authors study the learning of personalized reward models in contexts with high disagreement. They first analyze the reward learning problem using empirical risk minimization and provide a theoretical analysis of the generalization bound. Then, the authors propose to represent the reward model as the inner product between the shared user features and the user-specific feature. The reward model can be trained using empirical risk minimization on the pair-wise ranking data. When adapting to new users, only the new user-embedding vector needs to be updated with the shared user features fixed. The experiment is carried out on a synthetic dataset.

**Questions:**

Please refer to the Strengths And Weaknesses for questions. I also have the following question:

1. The authors mentioned the data in S is not i.i.d in line 126 on Page 3, could the authors explain how the proof addresses this challenge?

**Ethical Concerns:**

["NO or VERY MINOR ethics concerns only"]

**Final Justification:**

The authors' rebuttal addresses some of the concerns. However, my concerns about weakness points 3,4 and 6 still exist.

**Limitations:**

yes

**Quality:**

3

**Strengths And Weaknesses:**

Strengths:

1.	This paper studied an important problem of reward personalization, which can have a broad impact on LLM finetuning.

2.	To my knowledge, the theoretical analysis of reward personalization from the principle of empirical risk minimization is novel.

3.	The proposed personalized reward model can adapt to new users.

Weaknesses:

1.	One major concern is the scalability of the proposed reward model. In the proposed model, each user has a user-specific feature vector w_h. However, in practice, there can be a large number of users, which brings the scaling challenge.

2.	Another major concern is the capacity of the linear feature reward model. Considering the complexity of real users’ preferences, it is challenging to model using a linear combination of reward features.

3.	The connection between the theoretical analysis and the proposed reward-feature model is not tight. The theoretical analysis demonstrates that personalization is helpful to obtain a better bound. However, the importance of personalization has been widely recognized by the LLM personalization literature, also discussed in Section 6.

4.	The synthetic experiment is not convincing in demonstrating the superiority of the proposed reward-feature model. For example, in the synthetic dataset, there are only $2^{13}$ users and 13 features can be far from real user preferences. Given the concern about the capacity of the proposed reward, the authors need to use a more realistic dataset for evaluation.

5.	As the number of reward features is a key parameter in the proposed reward feature model, it’s better to evaluate the sensitivity of this hyperparameter.

6.	The baselines are weak. Considering many works on LLM personalization, more recent baselines are needed for comparison.

---

> ### Author Rebuttal · Authors · 2025-07-30
>
> Thank you very much for your thoughtful comments and detailed review! **We believe your review will be a great asset to accompany the paper, as it has discussions of some important aspects of our work in an extended (and slightly more informal) way.**
>
>
> &nbsp;
>
> ---
>
> > **1. In the proposed model, each user has a user-specific feature vector w_h. However, in practice, there can be a large number of users, which brings the scaling challenge.**
>
>
> &nbsp;
>
>
> Any model that has user-dependent parameters will face this challenge. With a single vector of coefficients per user, RFM is arguably at the most favorable end of this spectrum. As an illustration, in our case we used a reward model with 2 billion parameters, which is small for current standards. Using 128 features per user, we would need more than 15.6 million users to double the size of the model.
>
>
> Importantly, one of the reasons to split the set of parameters in two is that this allows for a distributed architecture in which user-specific parameters can be stored independently from the shared parameters (see discussion on line 196). For example, while the shared parameters $\theta$ should probably be stored in a fast-access, centralised memory, the vector $w_h$ could be delegated to a secondary memory and only be fetched on demand. Note that many of the current large-scale personalised software relies on similar architectures, with users described by local parameters that are often much more complex than a vector of coefficients.
>
>
>
> &nbsp;
>
> > **2. Considering the complexity of real users’ preferences, it is challenging to model using a linear combination of reward features.**
>
>
> &nbsp;
>
>
> Although the RFM approximation is linear in the features, the features themselves are arbitrary, potentially highly non-linear, functions of the inputs learned based on the preferences of raters sampled from the same distribution as future users. Therefore, with enough parameters and features, we can approximate the users’ preferences at any desired level of accuracy. In principle, if we have one feature per user and a large enough approximator, we effectively have a separate reward model per user. In practice, we only need the users’ preferences to lie on a $d$-dimensional manifold, where $d \ll |\mathcal{H}|$. As long as we have an approximator with at least $d$ features and enough capacity, we should be able to learn the geometry of this manifold given enough raters $m$ and examples per rater $n$, as indicated by our theoretical results (modulo optimisation issues).
>
> A non-linear dependency on the features could in some cases more easily capture user-specific reward structures. However, having a non-linear dependency on the features introduces complications. Most notably, it is unclear how to obtain a personalised LLM without having to go through the fine-tuning RL process for each user, which is unfeasible (unless one is willing to resort to methods without performance guarantees). In contrast, with an approximation that is linear in the features, one can use known RL techniques to synthesise a personalised LLM **without any learning involved and with performance guarantees [4–7, 28 , 43 , 11]**. A non-linear dependency on the features also gives rise to a non-convex adaptation, which usually requires more data and comes with weaker guarantees. See Response 6 to Reviewer LWzi for an illustration.
>
> Please refer to Responses 1 and 2 to Reviewer LWzi for related discussions.
>
>
> &nbsp;
>
>
> > **3. The connection between the theoretical analysis and the proposed reward-feature model is not tight. The theoretical analysis demonstrates that personalization is helpful to obtain a better bound. However, the importance of personalization has been widely recognized by the LLM personalization literature.**
>
>
> &nbsp;
>
>
> First, we would like to say that, although it is indeed true that the benefits of personalisation have been acknowledged in the LLM literature, the arguments are often at an intuitive (and sometimes empirical) level. We believe that rigorously quantifying these benefits is an important contribution to the ongoing research.
>
> Our theoretical results are indeed not specific to RFM. Note though that, as explained in the subsection “Connection with the theory” of Section 4, we can specialise them to provide guarantees on RFM _after adaptation_ (that is, on the final user). We accomplish that by folding in the adaptation process into the model’s mechanics, effectively removing the vector of coefficients $w$ from the set of parameters. Although we propose this “trick” mostly as an analytical device, it can in fact be used in practice, precisely because RFM’s architecture is linear in the features: the calculation of $w$ comes down to a convex optimisation that can be quickly solved. In contrast, an architecture non-linear in the features would require folding a non-convex optimisation into the mechanics of the model, which in general is not feasible.
>
>
> &nbsp;
>
>
> >  **4. [...] in the synthetic dataset, there are only  users and 13 features can be far from real user preferences. [...]the authors need to use a more realistic dataset for evaluation.**
>
>
> &nbsp;
>
>
> We highlight that we *do* have experiments with more realistic datasets. Please see results in Figure 2(c), whose data was generated by real (aggregated) human preferences. Furthermore, following a suggestion by Reviewer mJwX, we have carried out additional experiments with another dataset, PersonalLLM, which also contains data based on real human preferences (see Response 1 to Reviewer mJwX). In both cases RFM considerably outperformed the baseline, which represents the current practice.
>
>
> Although we agree that 13 features are probably not sufficient to capture human preferences in its full generality, we highlight that the resulting problem is far from trivial. To illustrate this, note that the in-context LLM solutions were unable to capture these features (cf. Figure 2(b)), even though they can capture simpler features (cf. Figure 8 and discussion on line 743).
>
> Please also see Response 3 to Reviewer 4gUj.
>
>
>
> &nbsp;
>
>
> > **5. As the number of reward features is a key parameter in the proposed reward feature model, it’s better to evaluate the sensitivity of this hyperparameter.**
>
>
> &nbsp;
>
>
> We note that we did vary the number $d$ of reward features in the experiments shown in Figure 1. Specifically, we tried $d  = 8, 32, 128$. As shown in the figure, $8$ features were not enough to fully capture the structure of the problem. Increasing $d$ to $32$ resulted in a significant boost in performance, but cranking it further up to $128$ did not help much. This makes sense, since in this case we know that the classification problem lies in a $13$-dimensional manifold. In all the experiments we ran, we noticed that, although sometimes increasing the number $d$ of features did not result in improved performance, it rarely (if ever) degraded the results.
>
>
> &nbsp;
>
>
> > **6. The baselines are weak. Considering many works on LLM personalization, more recent baselines are needed for comparison.**
>
>
> &nbsp;
>
>
> First, to make sure it does not go unnoticed, we highlight that we do have comparisons with _variational preference learning_ (VPL) [38], arguably one of the most prominent works in this area. The results we obtained with RFM were essentially on par with VPL’s, although our method is arguably much simpler (see Response 1 to Reviewer LWzi and Response 2 to Reviewer DjQY).
>
>
> Following a suggestion by Reviewer LWzi, we also carried out additional comparisons with the method presented in [ii], which uses non-linear features. Despite our honest attempts at improving the method’s performance, results are considerably worse than those of RFM. This provides further evidence in favour of having a simple, linear architecture on top of non-linear, learned features.
>
>
> Please also see Response 6 to Reviewer LWzi.
>
> [ii]. RLHF from heterogeneous feedback via personalization and preference aggregation.
>
>
> &nbsp;
>
>
> > **7. The authors mentioned the data in S is not i.i.d in line 126 on Page 3, could the authors explain how the proof addresses this challenge?**
>
>
> &nbsp;
>
>
>
> Thank you for the opportunity to discuss our theoretical results in a more informal way!
>
>
> In our setup, having iid data would mean first sampling a user $h \sim \mathcal{D}\_{\mathcal{H}}$, then sampling a context $x \sim \mathcal{D}^h_{\mathcal{X}}$, then sampling two responses $y, y’ \sim (\mathcal{D}^{h, x}\_{\mathcal{Y}})^2$, and finally sampling a preference $z \sim \mathcal{D}^{h, x, y, y’}\_{\mathcal{Z}}$. As explained on line 76, what we have instead is the sampling of $n$ tuples $t_h \coloneqq \{(x\_i, y\_i, y’\_i, z\_i) \}_{i=1}^n$ following each $h$. This data is non-iid, and so prevents directly applying classical results from learning theory. We circumvented this by changing perspective, and defining a new loss $\ell_n$ which is a function of $(c, h, t_h)$ rather than $(c, h, x, y, y’, z)$ (see equation 7). So, even though random variables defined via the original loss $\ell$ would not be  i.i.d., the variables $L^{c,n}$ we defined based on $\ell_n$ are (line 581).
>
>
> The challenge does not end there, however. Although we can apply the standard learning theory techniques to $L^{c, n}$, this will make $n$ “disappear” (that is, it will not appear in the bound). The way we elicited the dependency on $n$ was to do the following:
>
>
> - First, use Bernstein’s inequality (rather than the more commonly used Hoeffding’s counterpart) to expose the variance of $L^{c, n}$ (equation 8);
>
>
> - Then, use the law of total variance to decompose $\mathbb{V}(L^{c, n})$ into $\mathbb{E}[\mathbb{V}(L^{c, n}|H)]$ and $\mathbb{V}(\mathbb{E}(L^{c, n}|H])$ (equation 9);
>
>
> - Finally, note that $\mathbb{V}(L^{c, n}|H) = \dfrac{1}{n}\mathbb{V}(\ell|H)$ (equation 9).
>
>
> We hope this clarifies it!
>
> ---
>
>
> &nbsp;
>
>
>
> Thank you again! Let us know if any further clarifications are needed.

---

### Official Review · Reviewer_4gUj · 2025-07-01

**Clarity:** 3
**Significance:** 4
**Originality:** 3
**Rating:** 5
**Confidence:** 4

**Summary:**

This paper considers a more general scenario of Reinforcement Learning from Human Feedback (RLHF), addressing the potential user preference heterogeneity that is usually ignored in the conventional RLHF. This paper provides theoretical justifications showing that the estimate bias depends not only on the number of examples but also on the number of different raters. A new reward-model architecture, Reward-Feature Model (RFM), is introduced for personalized RLHF. RFM separates reward function’s parameters into shared ones and user-specific ones, enabling fast adaptation to new users by fine-tuning only user-specific parameters. Experiments with LLMs demonstrate RFM’s superiority over non-adaptive baselines, in-context personalization methods on the test accuracy in predicting user preferences.

**Questions:**

1. In real-world scenarios, users may not be able to provide 30 (or more)  high-quality samples. Have you tested performance under more extreme few-shot conditions (e.g., $m \leq 5$) and how does the model perform?
2. The bound established in Proposition 2 appears to exhibit strong dependence on the dimensions of $\mathcal{H}$, $\mathcal{X}$, and $\mathcal{Y}$. Is this bound tight or is it possible to further improve the dependence on dimensions?

**Ethical Concerns:**

["NO or VERY MINOR ethics concerns only"]

**Limitations:**

See weaknesses.

**Paper Formatting Concerns:**

The paper adheres to the formatting guidelines.

**Quality:**

4

**Strengths And Weaknesses:**

Strengths
1. This paper rigorously shows that how the estimate error depends on variance induced by user preference heterogeneity, and it can not be fully eliminated by only increasing the number of examples, which sheds the light on the necessity for considering user-specific reward models.
2. The proposed reward model, RFM, represent individual reward as a linear combination of  shared reward features, enabling data-efficient adaptation to new users with minimal feedback, which is well-suited for real-world scenarios where only limited preference data may be available for new users.
3. Empirical results show that RFM outperforms the non-adaptive baseline and demonstrates particularly strong advantages in scenarios with higher user heterogeneity. It also shows that RFM can rapidly adapt to new users only with a few examples. Experiments with LLM verify that RFM provides a good reward function for LLM to produce better response compares to the baseline. Comparisons based on the real-world dataset are also provided.

Weakness
1. Although Figure 1 demonstrate the model's effectiveness particularly under high user heterogeneity, it is worth noting that the synthetic data is constructed precisely following the RFM framework. It remains unclear whether the results showed in Figure 1 would generalize to preference data that does not conform to this modeling assumption.
2. The experimental validation with real human preference data only includes comparisons between RFM and the baseline model, omitting performance benchmarks against other adaptive methods.

---

> ### Author Rebuttal · Authors · 2025-07-30
>
> Thank you very much for your thoughtful comments and suggestions! We **have run an additional experiment to address your question regarding extreme few-shot adaptation conditions**. Following your comment, **we have also added an adaptive baseline to one of our experiments with real human preference data**.
>
> &nbsp;
>
>
> ---
>
> > **1. In real-world scenarios, users may not be able to provide 30 (or more) high-quality samples. Have you tested performance under more extreme few-shot conditions (e.g., m <= 5) and how does the model perform?**
>
> &nbsp;
>
> In the context of LLMs, providing a sample for adaptation means choosing between two candidate responses to a given prompt. The collection of sample responses does not have to happen all at once; for example, a system can present the user with alternative responses at reasonable intervals. Under these conditions, collecting 30 samples does not seem infeasible.
>
>
> That said, the reviewer is right to point out that in some circumstances 30 samples may be too many; also, regardless of the practical use of the model, understanding its behaviour under extreme few-shot conditions is a relevant scientific question. To investigate this question, we extended the RFM(32) results in Figure 1(c) to include test accuracies in 1-, 3-, and 5-shot settings. The results are shown in the table below. RFM(32)'s results in Figure 1(c) are also shown for reference.
>
>
> &nbsp;
>
> | $\hat{n}$ | Test accuracy +- 99% confidence interval|
> |---|---|
> | 1 | 0.5481 +- 0.0217 |
> | 3 | 0.5749 +- 0.0109 |
> | 5 | 0.5891 +- 0.0056 |
> | 10 | 0.7053 +- 0.0182 |
> | 30 | 0.7657 +- 0.079 |
> | 50 | 0.7770 +- 0.0333 |
> | 70 | 0.7835 +- 0.0217 |
> | 90 | 0.7900 +- 0.0172 |
>
>
> &nbsp;
>
> As expected, the model performance improves monotonically as the number of examples $\hat{n}$ used for adaptation increases. Note that even the results with a single example ($\hat{n}=1$) remain above random chance (though significantly lower than the 30-shot test accuracy).
>
>
> We point out that worse performance with very few adaptation examples is expected, for it reflects the intrinsic difficulty of the learning problem. Given that each user's preferences are derived from a combination of 13 features, many different combinations of those 13 features may explain the preference behind a few training examples, and multiple training examples may be necessary to disambiguate. It is reassuring to see that RFM is still able to capture some of the problem structure under these extreme conditions, and that its performance monotonically increases with $\hat{n}$.
>
> We plan to add these results to the paper in case future readers are also curious about RFM's performance under extreme few-shot conditions; please let us know if you have further suggestions.
>
> &nbsp;
>
> > **2. The bound established in Proposition 2 appears to exhibit strong dependence on the dimensions of H, X, and Y. Is this bound tight or is it possible to further improve the dependence on dimensions?**
>
> &nbsp;
>
> By assuming structure on either the hypothesis class or data distribution, tighter generalisation bounds can be obtained through e.g. Rademacher complexities; please see our Response 4 to Reviewer LWzi for how this looks in practice. Our aim with the bounds as displayed in the paper is to illustrate the form of generalisation bounds we obtain with our non-iid data, in particular its dependency on the number of raters $m$, the number of examples per rater $n$, and the inter- and intra-user loss variance terms in these bounds ($\mathbb{V}(\mathbb{E}[\ell_c |H])]$ and $\mathbb{E}[\mathbb{V}(\ell_c |H )]$). Instead of making assumptions about the hypothesis class or data distribution to derive the tightest possible bound, we chose to present a more general result that clearly shows the dependence of the generalisation error on the factors above.
>
> Please see Response 4 to Reviewer LWzi for an example of how our bound can be specialised using the Rademacher complexity.
>
>
> &nbsp;
>
> > **3. Although Figure 1 demonstrate the model's effectiveness particularly under high user heterogeneity, it is worth noting that the synthetic data is constructed precisely following the RFM framework. It remains unclear whether the results showed in Figure 1 would generalize to preference data that does not conform to this modeling assumption.**
>
>
> &nbsp;
>
> We call attention to the results shown in Figure 2(c), whose data was generated by real (aggregated) human preferences. Furthermore, following a suggestion by Reviewer mJwX, we have carried out additional experiments with another dataset, PersonalLLM, which also contains data based on real human preferences (see Response 1 to Reviewer mJwX). In both cases RFM considerably outperformed the baseline, which represents the current practice.
>
>
> We used features to generate the synthetic data mostly as a strategy to easily change the number of users and the heterogeneity of their preferences (the same strategy is suggested in [i]; see their equation 1). Although strictly speaking it is true that the resulting synthetic data conforms with the modelling assumption underlying our model, we highlight that RFM never had access to the “real” features.
>
>
> Finally, we argue that the modelling assumption underlying RFM should hold in many real scenarios. We can think of each user as inducing a different classification problem over the same inputs (that is, users only change the labelling of the inputs). In the limit, if we have at least as many features as users, and enough representational capacity, we effectively have a separate classifier per user. More realistically, we expect there to exist a $d$-dimensional manifold over which the classification problems induced by all the users are (approximately) linear. If this is the case, and as long as we have at least $d$ features and enough capacity, we will eventually learn such a manifold given enough raters $m$ and examples per rater $n$, as shown by our theoretical results (modulo the usual optimisation difficulties associated with deep learning).
>
>
> For related discussions, please also see Responses 2 and 4 to Reviewer DjQY and Responses 1 and 2 to Reviewer LWzi.
>
>
> [i]. PersonalLLM: Tailoring LLMs to Individual Preferences. Thomas P. Zollo, Andrew Wei Tung Siah, Naimeng Ye, Ang Li, Hongseok Namkoong.
>
>
> &nbsp;
>
>
> > **4. The experimental validation with real human preference data only includes comparisons between RFM and the baseline model, omitting performance benchmarks against other adaptive methods.**
>
>
> &nbsp;
>
>
> We have emphasised the comparison with the baseline in our experiments because it represents the current practice. But we agree that including the results from other adaptive models in experiments with real human preference data would provide a more comprehensive picture.
>
> To address this, we leveraged the new experiment we conducted on the PersonalLLM dataset, following a suggestion by Reviewer mJwX, which uses real human preference data. In addition to comparing RFM(128) with the baseline, we also compared it with the linear baseline (as described on line 326), **which is the adaptive model that performed best in our comparisons (cf. Figure 2(b) and Response 6 to Reviewer LWzi)**. The results are shown below:
>
> &nbsp;
>
> | Held-out model | Baseline | Linear baseline | RFM(128) |
> | :--- | :--------------- | :--------------- | :--------------- |
> | **0** | 0.3660 +- 0.0000 | 0.6534 +- 0.0013 | **0.7118 +- 0.0015** |
> | **1** | 0.8378 +- 0.0015 | 0.8438 +- 0.0005 | **0.8604 +- 0.0015** |
> | **2** | 0.8252 +- 0.0005 | 0.8252 +- 0.0005 | **0.8260 +- 0.0023** |
> | **3** | 0.8270 +- 0.0000 | 0.8444 +- 0.0006 | **0.8680 +- 0.0016** |
> | **4** | 0.8374 +- 0.0006 | 0.8260 +- 0.0008 | **0.8594 +- 0.0017** |
> | **5** | 0.7998 +- 0.0005 | **0.8190 +- 0.0000** | 0.8160 +- 0.0008 |
> | **6** | 0.8446 +- 0.0006 | 0.8428 +- 0.0005 | **0.8536 +- 0.0006** |
> | **7** | 0.6148 +- 0.0005 | 0.6060 +- 0.0000 | **0.6630 +- 0.0028** |
> | **8** | 0.6280 +- 0.0000 | **0.6422 +- 0.0005** | 0.6350 +- 0.0008 |
> | **9** | 0.6280 +- 0.0000 | 0.6022 +- 0.0005 | **0.6446 +- 0.0050** |
>
> &nbsp;
>
> **RFM outperforms the adaptive linear baseline in 8 out of 10 cases**. We conjecture that the results above are an underestimate of RFM’s performance in a real scenario. This conjecture stems from the observation that the reward models playing the role of users in this experiment reflect the preferences of aggregated real users. This means that some of the heterogeneity of the real population has probably been washed away in the training of these models. If the reward models were replaced by the real users underlying these models, RFM’s advantage over the two baselines would likely be greater still.
>
> We plan to add the results above to the paper in a figure analogous to Figure 2(c); please let us know if you have any further suggestions.
>
> Please see Response 1 to Reviewer mJwX for further details on the experiments above.
>
>
>
> ---
>
> &nbsp;
>
> We thank the reviewer again for their thoughtful review! Please let us know if you have any further suggestions to improve the paper.

---

### Official Review · Reviewer_LWzi · 2025-07-03

**Clarity:** 3
**Significance:** 3
**Originality:** 2
**Rating:** 4
**Confidence:** 4

**Summary:**

The paper highlights a key limitation in the current RLHF paradigm which is the assumption of homogeneous human preferences since users often disagree especially on subjective tasks hence learning a single, averaged reward model may misrepresent many individuals. To mitigate the above, the paper proposes an efficient framework for adaptive and personalized reward learning via learning shared features across users and with user-specific features for personalization. The paper provides strong theoretical guarantees with PAC bounds on the generalization error and empirical results showing benefit over standard (non-personalized) reward models.

**Questions:**

See Strengths And Weaknesses

**Ethical Concerns:**

["NO or VERY MINOR ethics concerns only"]

**Final Justification:**

I have read the rebuttal and most of my concerns are addressed.

**Limitations:**

See Strengths And Weaknesses

**Quality:**

3

**Strengths And Weaknesses:**

One of the key strengths of the paper lies in a rigorous theoretic analysis with PAC-style bounds that explicitly model inter-user and intra-user variance in reward modeling. This shows how the generalization error depends separately on the number of users, number of examples per user and variance across user-preferences. One key interesting concept that increasing data per user has diminishing returns unless the number of users also increases. The proposed approach generalizes to new users with as few as 10–30 labeled comparisons and thus efficient personalization is enabled without any need for gradient-based fine-tuning of the full model. Empirical results are nice which shows that the approach outperforms single utility, non adaptive, in-context based baselines.

Although the paper provides a principled approach, however the proposed method still assumes linear parametrization which may be overly restrictive in practice and fail to capture complex user-specific reward structures. Additionally, the feature $\phi$ requires futher assumptions? While adapting to new user preferences, which is a key component to the approach, how can we ensure $\phi$ will generalize to them ? The paper does not address what assumptions on user preference distributions or feature richness are required to guarantee that $\phi$ will generalize well to unseen users.

The paper provides strong theoretical guarantees, however the PAC bounds are standard and the paper doesn't explain why or what are the exact challenges or unique components in the current theoretical results for the specific problem? The bounds use covering number, but do not consider Rademacher complexity or VC-dimension, which could provide sharper insights or tighter sample complexity results.

The term Var(E[l_c|H)] plays an extreme crucial role in characterizing the diversity, which play a critical role in quantifying preference heterogeneity. However, the interpretation of these terms is only briefly discussed, and a more detailed explanation is essential. In particular, how these terms scale with respect to population size, user diversity, or feature misalignment should be discussed to better understand when the proposed method will be effective.

Finally, the experimental section needs comparison with more SoTA baselines with multi-rewards which is currently missing. It will be helpful to compare and contrast with [1], [2] which are related and highlight the key benefits over the above methods. This would help position the proposed approach more clearly in relation to existing work and highlight its empirical advantages or trade-offs.


[1]. Provable Multi-Party Reinforcement Learning with Diverse Human Feedback
[2]. RLHF from heterogeneous feedback via personalization and preference aggregation

---

> ### Author Rebuttal · Authors · 2025-07-30
>
> Thank you for your thoughtful comments! Following your suggestions, **we've derived an additional theoretical result based on the Rademacher complexity and carried out an extra experiment featuring the SOTA comparisons you recommended**. We are confident these additions significantly enhance the paper's quality.
>
> &nbsp;
>
> ---
>
> >  **1. [...] the proposed method still assumes linear parametrization which may be overly restrictive in practice and fail to capture complex user-specific reward structures.**
>
> &nbsp;
>
> Although the RFM approximation is linear in the features, the features themselves are arbitrary, potentially highly non-linear, functions of the inputs. They are also learned based on the preferences of raters sampled from the same distribution future users will come from. This means that, with enough parameters and features, we can approximate the users’ preferences at any desired level of accuracy. In principle, if we have one feature per user and a large approximator, we effectively have a separate reward model per user. In practice, we only need the users’ preferences to lie on a lower-dimensional manifold, and have enough raters and data in the training set to capture the geometry of this manifold.
>
> That said, we acknowledge that a non-linear dependency on the features could in some cases more easily capture user-specific reward structures (in the sense that we would need fewer user-specific parameters). We do think however that the benefits provided by a linear dependency on the features largely outweigh this potential drawback. These benefits are discussed on line 249, and summarised here:
>
> 1. The adaptation problem is convex;
>
> 2. We can use robust linear algebra techniques to improve the adaptation [46];
>
> 3. The features (and their contributions) are interpretable;
>
> 4. We can add handcrafted features;
>
> 5. Importantly: **we can resort to known RL techniques to synthesise a personalised LLM based on the vector of coefficients $w$, without any learning involved and with performance guarantees [4–7, 28 , 43 , 11]**. With a non-linear dependency on the features, one has to either fine-tune the LLM with RL for each user, which is impractical, or resort to methods without performance guarantees.
>
> Please also see Response 2 to Reviewer DjQY.
>
>
> &nbsp;
>
> > **2. Additionally, the feature  requires futher assumptions? While adapting to new user preferences, which is a key component to the approach, how can we ensure  will generalize to them ?**
>
> &nbsp;
>
> We can think of each user as a separate classification problem over the same input space. Under this view, we have to learn features such that all labellings of the inputs induced by the users are a linear classification. In other words, users’ preferences lie in a $d$-dimensional manifold, and our goal is to learn such a manifold. As long as we have an approximator with at least $d$ features and enough capacity, we should be able to learn the manifold given enough raters $m$ and examples per rater $n$, as indicated by our theoretical results (modulo standard optimisation issues). We will add this explanation to the paper.
>
> Please see Response 3 to Reviewer 4gUj for a related discussion.
>
> &nbsp;
>
> > **3. …the PAC bounds are standard and the paper doesn't explain why or what are the exact challenges or unique components in the current theoretical results for the specific problem?**
>
> &nbsp;
>
> One challenging aspect of our theoretical analysis, mentioned on lines 44 and 126, is the fact that the data is not i.i.d. Another interesting aspect is the use of Bernstein’s concentration inequality instead of the more commonly used Hoeffding's counterpart. This was necessary to surface the dependency of the generalisation error on the inter- and intra-user loss variances, as noted in your review.
>
> Please see Response 7 to Reviewer DjQY for an extended discussion.
>
> &nbsp;
>
> > **4. The bounds use covering number, but do not consider Rademacher complexity or VC-dimension, which could provide sharper insights or tighter sample complexity results.**
>
> &nbsp;
>
> The main goal of our theoretical analysis is to show the qualitative dependency of the generalisation error on $n$, $m$, and the inter- and intra-user loss variances. Instead of making structural assumptions to get the tightest possible bound, we chose to present a general result that illustrates the relevant trends.
>
> That said, the point raised by the reviewer is interesting. It is indeed possible to obtain stronger results in our setup when assuming additional structure on the hypothesis class. We briefly indicate how Rademacher complexity bounds, as the reviewer suggests, can be used to obtain such bounds when the classifier is logistic regression over concatenated embeddings of user, prompt, and responses.
>
> This bound can be obtained by following a standard Rademacher generalisation bound argument (e.g. as in Chapter 26 of [45]) at the level of user data, which relies on McDiarmid's inequality, and then applying Bernstein's inequality later in the argument to reveal a dependence on the $n$ parameter, as well as inter- and intra-user loss variances. Such an argument yields the following bound:
>
> $$
>     L_\mathcal{D}(c^\*) \leq L_\mathcal{D}(\tilde{c}^\*) + \frac{2W}{\sqrt{m}} + 3 \sqrt{\frac{g}{2m}} + \frac{1}{3m}\Bigg[ g + \sqrt{g^2 + 18gm\big( \frac{1}{n} \mathbb{E}[\mathbb{V}(\ell_c |H )] + \mathbb{V}(\mathbb{E}[\ell_c |H])] \big)} \Bigg],
> $$
>
> where $g=\log(6/\delta)$. Here, we assume the concatenation of embeddings is guaranteed to have norm at most 1, and that we consider fitting weight vectors in the logistic regression of norm at most W, as is common in the analysis of such regression models.
>
> We will add the above result to the paper; thank you for your suggestion!
>
> &nbsp;
>
> > **5. The term Var(E[l_c|H)] plays an extreme crucial role [...] In particular, how these terms scale with respect to population size, user diversity, or feature misalignment should be discussed to better understand when the proposed method will be effective.**
>
> &nbsp;
>
> Thank you for the question; we agree that the term $\mathbb{V}(\mathbb{E}[\ell_c |H])$ plays a crucial role. We will add a more detailed discussion. Here's a brief summary:
>
> $\mathbb{V}(\mathbb{E}[\ell_c |H])$ depends both on the diversity of the population $\mathcal{H}$ and the capacity of the classifier $c$ to capture it. Therefore:
>
> - **User diversity**: for a fixed classifier $c$,  $\mathbb{V}(\mathbb{E}[\ell_c |H])$ should never decrease with user diversity, and it will in general increase.
>
> - **Feature misalignment**: For a fixed population $\mathcal{H}$,  $\mathbb{V}(\mathbb{E}[\ell_c |H])$ should never decrease with more misaligned features, and it will in general increase.
>
> - **Population size**:  $\mathbb{V}(\mathbb{E}[\ell_c |H])$ does not necessarily increase with the population size (if all users $h$ have the same preferences, this term is zero regardless of $|\mathcal{H}|$). In practice, we expect $\mathcal{H}$ to generally become more heterogeneous with its size, which likely yields an increase in  $\mathbb{V}(\mathbb{E}[\ell_c |H])$, as discussed.
>
> &nbsp;
>
> > **6. It will be helpful to compare and contrast with [i]. Provable Multi-Party Reinforcement Learning with Diverse Human Feedback [ii]. RLHF from heterogeneous feedback via personalization and preference aggregation**
>
> &nbsp;
>
> We have carried out a comparison between RFM and [ii]. We chose [ii] because their method is slightly more general than [i]. Using our terminology, we can describe [ii] as having a single learning phase that is intermediate between training and adaptation. If we partition the shared parameters $\theta = [\theta_1, \theta_2]$, the method consists in keeping $\theta_1$ fixed while $\theta_2$ is learned together with the vectors $w_h$ associated with users $h \in \mathcal{H}$.
>
> Following the experiments in [ii], we achieved this by freezing the backbone Gemma model ($\theta_1$), representing $\phi$ as a multilayer perceptron (MLP) with $k$ layers ($\theta_2$), and learning it together with $w_h$ (to make sure $\theta_2$ was initialised with reasonable values, we re-ran our training phase with both $\theta_1$ and $\theta_2$ being learned). We then repeated the experiments shown in Figure 2(b) using this architecture. The results for $\phi$ using $k=3$ and $k=5$ hidden layers with 32 units each can be seen in the table below. For reference, we also included the other results in Figure 2(b).
>
> &nbsp;
>
> | Model                     | $\hat{n}$ | Test accuracy +- 99% confidence interval |
> | :------------------------ | :-------- | :-------- |
> | Baseline                  | 10        | 0.6258 +- 0.0246                         |
> | Linear baseline           | 10        | 0.6285 +- 0.0176                         |
> | Non-linear $\phi$ (3 layers) | 10        | 0.5543 +- 0.0381                         |
> |                           | 100       | 0.5706 +- 0.0207                         |
> | Non-linear $\phi$ (5 layers) | 10        | 0.5237 +- 0.0273                         |
> |                           | 100       | 0.5620 +- 0.0452                         |
> | RFM(32)                   | 10        | 0.7053 +- 0.0182                         |
>
>
>
> &nbsp;
>
> The models with non-linear $\phi$ do not perform well. We conjectured that this may be due to the number of parameters to be learned, so we re-ran the experiment using $\hat{n} = 100$ examples rather than $10$. Although the results improved slightly, they are still below the baselines, and considerably below RFM.
>
> We will include these results in Figure 2(b).
>
> As a final note, we highlight the fact that we have comparisons with _variational preference learning_ (VPL) [38] –arguably also a SOTA method–, as described in the subsection “Comparisons” of Section 5.
>
> Please see Response 6 to Reviewer DjQY for a related discussion.
>
> ---
>
> &nbsp;
>
> Thank you again for your review! Please let us know if there are any further concerns to be addressed.

---

> > ### Author Response · Authors · 2025-08-06
> > **Follow-up**
> >
> > Thank you again for your detailed review! We believe our rebuttal addresses all the questions raised. This includes deriving an additional theoretical result based on the Rademacher complexity, as suggested by you, and also running an extra experiment with one of the baselines you recommended.
> >
> > If you feel satisfied with our responses, we kindly request that you reconsider your recommendation to reject our paper. If, on the other hand, you have lingering concerns, please let us know and we will be happy to discuss further.

---

### Official Review · Reviewer_mJwX · 2025-07-07

**Clarity:** 3
**Significance:** 3
**Originality:** 4
**Rating:** 5
**Confidence:** 3

**Summary:**

The authors formalize the problem of user-specialized reward modeling for RLHF in settings with heterogeneous human preferences, deriving a PAC bound that highlights the dependence of approximation error on both data quantity and rater diversity. They propose an adaptive reward model where individual preferences are represented as a linear combination of shared reward features, enabling fast personalization to new users with few examples.

**Questions:**

Can you conceptually compare your approach with LoRe (https://arxiv.org/pdf/2504.14439)? It appears to be highly relevant to your work.

**Ethical Concerns:**

["NO or VERY MINOR ethics concerns only"]

**Final Justification:**

With the new results added during the rebuttal, I believe this paper will make a positive contribution to our community, and I am raising my score to 5.

**Limitations:**

yes

**Paper Formatting Concerns:**

null

**Quality:**

3

**Strengths And Weaknesses:**

**Strengths**
- Overall idea: The paper addresses a key limitation in current RLHF for LLMs by introducing user-specialised reward models that adapt to individual preferences, including those of users not present in the training data. The idea that individual preferences can be represented as a linear combination of shared reward features is both intuitive and practically implementable.
- Reproducibility: Although training scripts are not released due to proprietary constraints, the paper provides comprehensive details on experimental setups, model configurations, hyperparameters, and adaptation procedures, making reproduction of the main results feasible. It would be helpful to improve the reproducibility if the authors can release the rater annotations.
- Paper presentation: The paper is clearly written and well-structured, presenting complex theoretical analyses and empirical methods in a logical progression. The formalization of the problem and the derivation of PAC bounds are particularly clear, facilitating understanding of both the motivation and the solution.

**Weaknesses**
- The experimental evaluation relies primarily on the UltraFeedback dataset with synthetic raters, which, while offering high experimental control, limits the demonstration of the model’s effectiveness on diverse datasets. The empirical claims would be strengthened if the authors can show the results on the estabished preference datasets for personlization like personalLLM (https://arxiv.org/abs/2409.20296).

---

> ### Author Rebuttal · Authors · 2025-07-30
>
> Thank you for your comments, and in particular for calling our attention to both the PersonalLLM dataset and the LoRe work! Following your suggestion, **we carried out an additional experiment using the PersonalLLM dataset**. We also like your suggestion to make our rater annotations available, and will follow it upon the release of the paper.
>
>
> &nbsp;
>
> ---
>
> > **1. The empirical claims would be strengthened if the authors can show the results on the estabished preference datasets for personlization like personalLLM.**
>
> &nbsp;
>
> Thanks again for pointing us to the PersonalLLM dataset, it is indeed a great asset in the study of adaptive reward models. Leveraging the fact that the examples in PersonalLLM have been scored by reward models, we conducted an experiment analogous to the one described in the subsection “Modelling groups of real users” of our paper’s Section 5.
>
> We now briefly describe the details of the experiment and show the results. Each prompt in the PersonalLLM dataset has 8 responses, and each response has been scored by 10 reward models. For our experiments, we picked the first two responses to each prompt $x$, and used them as our pair $(y, y’)$. We then proceeded as in the experiments described in the section “Modelling groups of real users”: we filtered out all the examples in the training and test sets in which $2$ or fewer raters disagreed with the majority, and then performed leave-one-out cross-validation, holding out each model in turn as a user while using the remaining 9 models as training raters.
>
> Below we show the results obtained by the baseline, the linear baseline (as described on line 326), and RFM(128). Each row shows the models’ accuracy in predicting the preferences of the held-out reward model (and corresponding 99% confidence intervals over $5$ runs).
>
> &nbsp;
>
> | Held-out model | Baseline | Linear baseline | RFM(128) |
> | :--- | :--------------- | :--------------- | :--------------- |
> | **0** | 0.3660 +- 0.0000 | 0.6534 +- 0.0013 | **0.7118 +- 0.0015** |
> | **1** | 0.8378 +- 0.0015 | 0.8438 +- 0.0005 | **0.8604 +- 0.0015** |
> | **2** | 0.8252 +- 0.0005 | 0.8252 +- 0.0005 | **0.8260 +- 0.0023** |
> | **3** | 0.8270 +- 0.0000 | 0.8444 +- 0.0006 | **0.8680 +- 0.0016** |
> | **4** | 0.8374 +- 0.0006 | 0.8260 +- 0.0008 | **0.8594 +- 0.0017** |
> | **5** | 0.7998 +- 0.0005 | **0.8190 +- 0.0000** | 0.8160 +- 0.0008 |
> | **6** | 0.8446 +- 0.0006 | 0.8428 +- 0.0005 | **0.8536 +- 0.0006** |
> | **7** | 0.6148 +- 0.0005 | 0.6060 +- 0.0000 | **0.6630 +- 0.0028** |
> | **8** | 0.6280 +- 0.0000 | **0.6422 +- 0.0005** | 0.6350 +- 0.0008 |
> | **9** | 0.6280 +- 0.0000 | 0.6022 +- 0.0005 | **0.6446 +- 0.0050** |
>
> &nbsp;
>
> As one can see, results are similar to those shown in Figure 2(c). We first focus on the comparison between RFM and the baseline, which represents the current practice. Our model outperforms the baseline in all cases. The baseline performs well in predicting some of the held-out model’s preferences, which probably means that these specific models “agree” with the majority. In these cases RFM still outperforms the baseline, although the difference is small. However, for other reward models (notably model number 0) the baseline performs poorly, likely as a consequence of the fact that these models disagree with the majority. In such cases RFM significantly outperforms the baseline. As noted in our previous experiments with real preference data, RFM can work as a form of “safety net” to make sure that minority preferences are also represented.
>
> Because the linear baseline can adapt to new users (albeit using user-agnostic features), it performs better than the baseline in most cases. Still, RFM outperforms the linear baseline in 8 out of 10 held-out models.
>
> We note that the reward models playing the role of users in this experiment in fact reflect the preferences of aggregated real users. This means that some of the heterogeneity of the real population has probably been washed away in the training of these models. We conjecture that if the reward models were replaced by the real users underlying these models, RFM’s advantage over the two baselines would be greater still.
>
> We plan to add the results above to the paper in a figure analogous to Figure 2(c); please let us know if you have any further suggestions.
>
> As a final note, we observe that, in addition to carrying out the experiments above, we have also conducted experiments with additional baselines, as described in Response 6 to Reviewer LWzi.
>
>
> &nbsp;
>
> > **2. Can you conceptually compare your approach with LoRe?**
>
> &nbsp;
>
> Thank you for calling our attention to the LoRe paper, we were not aware of it. One of the reasons we might have missed it is that it only became available on arXiv on 20 April, 2025 –that is, less than a month before NeurIPS’ deadline. As a kind reminder of NeurIPS’ policy, _“Papers appearing online after March 1st, 2025 are generally considered concurrent to NeurIPS submissions. Authors are not expected to compare to those.”_
>
> However, the reviewer is asking us to make a conceptual comparison, which seems fair. The comparison can be made at different levels of abstraction. From a strictly mechanistic perspective, the LoRe architecture is very close to ours, in that both our approaches express individual reward functions as linear combinations of learned reward features (or "reward basis" in LoRe's terminology). A small, but potentially relevant difference is that their vector of coefficients $w$ is a distribution (that is, $0 \le w_i \le 1$ and $\sum_i w_i = 1$). This restricts the expressiveness of the model.
>
> In terms of collecting empirical evidence in favour of the proposed architecture, Bose et al. make some of the same comparisons we do, namely: baseline, linear baseline, and VPL. They also compare LoRe with Chen et. al’s [10] _pluralist alignment_ (PAL), which we do not. On the other hand, they do not compare their method against the in-context LLM solutions, as we do in Figure 2(c), or with the non-linear features added following a suggestion by Reviewer LWzi (see Response 6 to that reviewer). They also use 3 datasets (PersonalLLM, Reddit TLDR, and PRISM), all of them considerably smaller than the UltraFeedback dataset we adopted (as discussed in Response 1 above, we have now also conducted experiments using PersonalLLM). Finally, Bose et al. do not investigate how LoRe’s performance transfers to the steering of the downstream LLM. As shown in Figure 2(a), we do so using best-of-$N$.
>
> At a more analytical level, Bose et al. do not present theoretical results. As discussed in Response 3 to Reviewer DjQY, our theoretical analysis sheds light on the learning of the features $\phi$. Specifically, we have shown how the associated generalisation error depends on the number of raters $m$, the number of examples per rater $n$, and the variance terms $\mathbb{E}[\mathbb{V}(\ell_c |H )]$ and $\mathbb{V}(\mathbb{E}[\ell_c |H])]$. That is, we have quantified how fast our approximation approaches the best possible approximation in its class.
>
> Besides providing a solid theoretical ground for RFM (and LoRe), our results also help to frame the learning of $\phi$ in the more rigorous terms of empirical risk minimisation. This in turn informs us how to think about the use of our model in practice. For example, we make it explicit an assumption that is usually only implicit in previous work: users and raters come from the same distribution. We also show in Figure 4 what happens when this is not the case.
>
> All that said, Bose et al.’s paper is a solid, well-executed piece of research. We see their concurrent work and ours as corroborating each other in building theoretical arguments and empirical evidence in favour of the proposed architecture. We will add a discussion on LoRe to our Related Work section as concurrent work.
>
> &nbsp;
>
> > **3. It would be helpful to improve the reproducibility if the authors can release the rater annotations.**
>
> &nbsp;
>
> This is a great idea! We will release the annotations along with the camera-ready version of the paper.
>
> ---
>
> &nbsp;
>
> We thank the reviewer again for their comments and suggestions! Please let us know if any further clarifications are needed, and also if you have any further suggestions.

---

> > ### Comment · Reviewer_mJwX · 2025-08-06
> >
> > Thanks for the detailed reply!  The authors' rebuttal addressed my major concerns (can be even better with a more concise rebuttal). With the solid new results, I believe the alignment community will benefit from the study. I remain positive about this work. Good luck.

---

### Note · Authors · 2025-08-11

## Rebuttal summary

---

We have made a genuine effort to properly address all the points raised in the reviews. Here’s a summary of our rebuttal:

- We conducted **an additional experiment comparing our proposed architecture, RFM, with a suggested baseline** from the current literature.

- We performed **a new experiment using the publicly available PersonalLLM dataset**. This experiment included an adaptive baseline to provide a direct comparison between RFM and an adaptive counterpart on a dataset with real human preference data.


- We performed **an experiment in which RFM was adapted to new users under extreme few-shot conditions (1, 3, and 5 examples only)**.

- We derived **a new PAC bound based on the Rademacher complexity** to strengthen our theoretical results.

- We clarified all questions regarding our theoretical results and proposed architecture. Regarding the latter, we **clarified that RFM’s linear dependency on features does not restrict its expressiveness, as the features themselves are arbitrary functions of the inputs**. This linear dependency is, in fact, the key to RFM’s desirable properties (see below).

We argue that this is a strong rebuttal. We thank Reviewer mJwX for re-stating their support after the rebuttal, and we interpret the lack of explicit responses from the other reviewers as a sign that all their concerns and questions have been properly addressed. We thank all the reviewers for their thoughtful feedback and kindly ask them to consider improving their scores.

&nbsp;

## Primary contributions

---

As a reminder, our primary contributions are:

- **A formal study of the problem of learning an adaptive reward model**. Our analysis provides a formal justification for the intuition that an adaptive reward model should be beneficial when there is disagreement among users.

- **The proposal of RFM, a simple adaptive architecture that adapts to new users with very few examples (~30 in our experiments)**. RFM's linear dependency on features yields a convex adaptation problem and allows for the synthesis of a personalised LLM without any further learning.

- **Our commitment to release the rater annotations used** to facilitate future research and comparisons.

We are confident that our work is a solid contribution, and the reviewers’ comments have been invaluable in improving it further.

&nbsp;

We kindly ask the AC and the reviewers to take all of the above into account when making a decision regarding our paper.

---

### Decision · Program_Chairs · 2025-09-17

**Decision:**

Accept (poster)

**Comment:**

This paper studies user-specialized reward modeling for RLHF under heterogeneous human preferences, deriving a PAC bound that links approximation error to data quantity and rater diversity. They propose an adaptive reward model that represents individual preferences as linear combinations of shared reward features, allowing rapid personalization to new users with minimal examples. It reached a consensus that the problem being considered is of practical significance, with solid theoretical analyses and experimental results. The manuscript is also well-written. There were some concerns regarding the representativeness of the datasets, comparison with existing closely-related baselines [1][2] (as suggested by Reviewer LWzi), and the tightness of the PAC bounds. These concerns have been adequately addressed by the authors' new experiments and bounds derived during the rebuttal. Overall, this paper clears the bar for acceptance at NeurIPS. I suggest the authors incorporate the feedback and the rebuttal (e.g., detailed and careful comparison with [1][2] and new results) in preparing the camera-ready version of the paper.


[1] Provable Multi-Party Reinforcement Learning with Diverse Human Feedback

[2] RLHF from heterogeneous feedback via personalization and preference aggregation